# Online Learning Of Neural Computations From Sparse Temporal Feedback

**Lukas Braun**[1,2]
lukas.braun@psy.ox.ac.uk

**Tim P. Vogels**[3]
tim.vogels@ist.ac.at

## Abstract

Neuronal computations depend on synaptic connectivity and intrinsic electrophysiological properties. Synaptic connectivity determines which inputs from presynaptic neurons are integrated, while cellular properties determine how inputs are filtered over time. Unlike their biological counterparts, most computational approaches to learning in simulated neural networks are limited to changes in synaptic connectivity. However, if intrinsic parameters change, neural computations are altered drastically. Here, we include the parameters that determine the intrinsic properties, e.g., time constants and reset potential, into the learning paradigm. Using sparse feedback signals that indicate target spike times, and gradient-based parameter updates, we show that the intrinsic parameters can be learned along with the synaptic weights to produce specific input-output functions. Specifically, we use a teacher-student paradigm in which a randomly initialised leaky integrate-and-fire or resonate-and-fire neuron must recover the parameters of a teacher neuron. We show that complex temporal functions can be learned online and without backpropagation through time, relying on event-based updates only. Our results are a step towards online learning of neural computations from ungraded and unsigned sparse feedback signals with a biologically inspired learning mechanism.

## 1 Introduction

In biological and artificial spiking neurons, information processing depends on synaptic connectivity as well as intrinsic variables. How a neuron integrates synaptic inputs over time changes its input-output function, leading to a large variety of neural computations [1]. In biology, the input-output function of a neuron is modified not only by synaptic plasticity [2], but also by alterations of the neuron's electrophysiology, e.g., through changes in the composition of trans-membrane ion channels, so-called *intrinsic plasticity* [3, 4]. Synaptic interactions influence the amount of neurotransmitter released from the presynaptic neuron or the amount of receptors in the postsynaptic terminal and therefore how strongly neurons are connected to each other. Intrinsic properties of the neuron, on the other hand, affect the neuron's threshold and resting potential and how strongly postsynaptic potentials are amplified [4].

Correspondingly, the computational properties of an artificial neuron depend on the values of its synaptic weights and intrinsic parameters. For instance, the neural computation performed by a leaky integrate-and-fire neuron [5] or a leaky resonate-and-fire neuron [6] with identical synaptic inputs and synaptic weights but different intrinsic parameters varies strongly within and across neuron models (Figure 1A). Learning in artificial neurons is thus the process of modifying a set of synaptic weights and intrinsic parameters to reach a target input-output function. However, unlike their biological counterparts, most computational approaches to learning in spiking neurons are limited to effects of

---

1. Department of Experimental Psychology, University of Oxford, Oxford, United Kingdom
2. Bernstein Center for Computational Neuroscience Berlin, Berlin, Germany
4. Institute of Science and Technology Austria, Klosterneuburg, Austria

35th Conference on Neural Information Processing Systems (NeurIPS 2021).

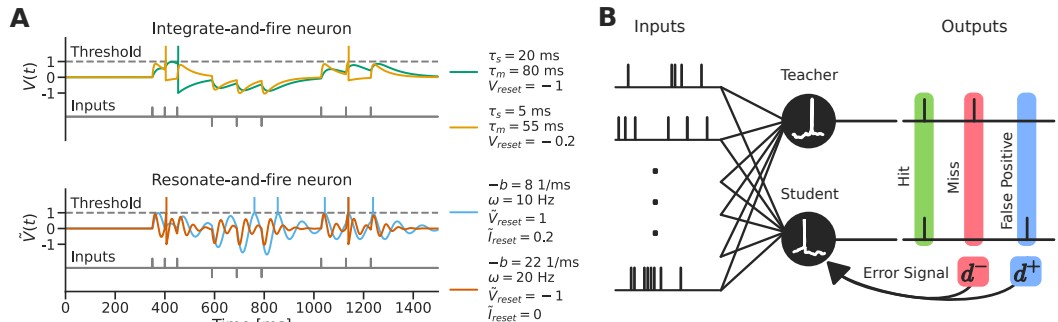

Figure 1: Neural computations and the teacher-student paradigm. A: Despite identical synaptic weights and synaptic inputs, the input-output function varies strongly within and across neuron models due to different intrinsic parameters. B: The teacher-student paradigm is used to study the convergence properties of a learning algorithm. The teacher and the student receive identical inputs, but their synaptic weights and intrinsic parameters have different values and therefore generate different outputs. The learning problem for the student neuron is to recover the fixed set of parameters of the teacher and thus to learn the neural computation performed by the teacher neuron.

alterations in synaptic connectivity. Here, we derive a gradient-based online learning rule for leaky integrate-and-fire neurons and leaky resonate-and-fire neurons that can learn the synaptic connectivity along their intrinsic parameters from sparse temporal feedback in the teacher-student paradigm.

## 1.1 Related Work

Applying gradient-based optimisation and the backpropagation algorithm [7, 8] to spiking neurons is difficult since the gradient of spike trains is ill-defined: it is zero everywhere except at spike times, where it is infinite. Methods to overcome the problem of non-differentiable spike trains include using gradients with respect to the probability density of spike-times [9], minimising the temporal distance between output and target spike times [10], temporal smoothing of synaptic-currents [11] or the discrete error signals caused by target-spikes [12, 13, 14] or by temporal gradient accumulation using the adjoint method [15]. In contrast to approaches that are directly optimising output spike times, there are also approaches that optimise the activity of spiking neurons through their network activity [16, 17], firing rate response function [18] or via the auxiliary objective of generating a fixed amount of output spikes for certain input spike patterns [19, 20]. Alternatively, so-called "surrogate gradient" methods replace the non-differentiable threshold function with a smooth surrogate function during the backpropagation of errors [21, 22, 23]. Using surrogate gradients and backpropagation through time [24], multilayer spiking neural networks can be successfully trained on a wide range of supervised learning tasks [25, 26, 27], even when errors are backpropagated at spike times only [28].

Our learning rule is closely related to the work by Gütig and Sompolinsky [19] and Memmesheimer et al. [29]. The tempotron learning rule [19] trains leaky integrate-and-fire neurons to emit an output spike for some input spike patterns, but to remain silent for others. Memmesheimer et al. [29] adapted this learning rule to learn synaptic weights that map input spike trains to a target output spike train. In contrast to the tempotron learning rule, the feedback signal does not indicate the class membership of an input spike pattern but target spike times: on the occurrence of each missed or false output spike the distance to the spiking threshold is minimised or respectively, maximised. If an output spike does not fall within a tolerance window around a target spike, a single parameter update is performed and the training procedure and neuron model are reset to the beginning of the training sequence. Consequently, the target spike train is learned one by one, from the first to the last spike of the target sequence, eventually requiring many replays of the sequence until the algorithm converges.

We extend the learning rule by Memmesheimer et al. [29] by an event-dependent scaling factor, such that synaptic weights and intrinsic parameters can be learned mutually and online, i.e., from a continuous stream of inputs without resetting the training procedure after each error and without relying on a tolerance window. Like Memmesheimer et al. [29], we train without backpropagating errors through time [24]. We hypothesise that a learning signal that indicates when a neuron should spike, paired with a gradient-based online parameter update rule, is sufficient to learn the complex temporal input-output relationships of a neuronal computation.

## 2 Methods

### 2.1 Teacher-student paradigm

Learning in artificial neurons relies on finding the set of parameters that facilitates a certain target computation. One approach to test an algorithm's learning capability is to test its ability to map randomly sampled input spike patterns to randomly sampled output spike trains (e.g. used in [12, 13, 23] and in [30] to compare algorithms). However, under this testing paradigm, the difficulty of the task is directly dependent on the statistics (e.g. dimensionality and firing rate) of the input spike patterns and target spikes. Therefore, the size and shape of the solution space and thus the difficulty of the task are unknown and strongly fluctuate with the choice of hyperparameters. As a consequence, the set of parameters that solves the problem may be large and thus even suboptimal learning algorithms can still yield good performance. For example, some of the solutions to a learning task may not be anticipated, like networks with biologically implausible high firing rates [27]. Further, using random input-output spike patterns of fixed length is unsuitable for evaluating an online-learning algorithm since, as the length of the random pattern increases, the probability of a solution existing is vanishing to zero.

The teacher-student paradigm [31, 32] circumvents this problem by generating target spike times using a model (i.e. the teacher neuron). The learning task is then to recover the teacher's parameter that determine its neural computation with a student neuron (e.g. in [29] "Reconstruction of Synaptic Weights"). Like this, the learning problem is well defined, as a single point in parameter space. In our case, learning signals are derived from comparing the output spike train of the teacher neuron with the output spike train of the student (Figure 1B). The teacher-student paradigm has several advantages, e.g., over the test performance in a standardised supervised learning task. First, if the teacher and student neuron have the same architecture, it is guaranteed that a unique solution to the training problem exists. Second, the student's parameters, and thus performance, can be compared to the solution during and after the training process. Finally, since the teacher is used to generate training samples, an infinite amount of unique and continuous training data can be sampled. In summary, by controlling the solution space of the task, the teacher-student paradigm can be used to precisely test the capability of performing spike-pattern recognition and the validity of assumptions of a learning algorithm.

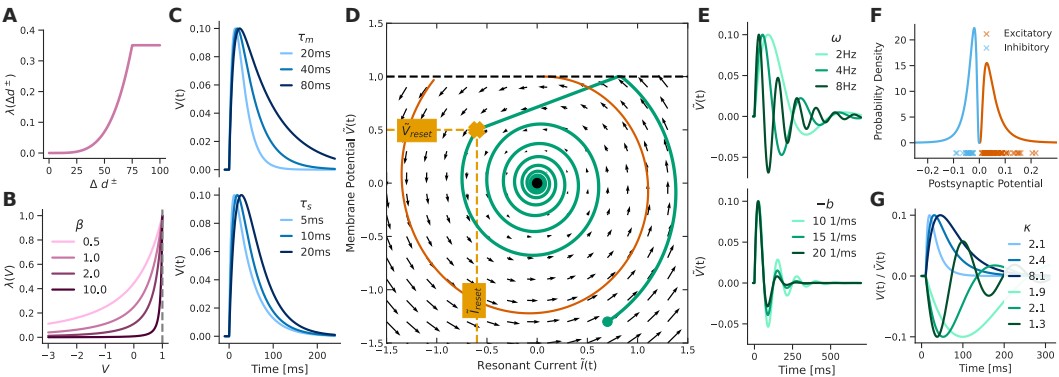

Figure 2: Methods. A: Event-dependent scaling factor. B: The SuperSpike [23] surrogate gradient for different $\beta$ values. C: Influence of the membrane ($\tau_m$) and synaptic time constant ($\tau_s$) on the membrane potential. While the former determines how quickly the membrane potential falls back to rest, the later shifts the position of the peak amplitude. D: Phase plane of the leaky resonate-and-fire neuron with example trajectory (green), separatrix (orange) and post spike reset position. E: Influence of the frequency of the subthreshold potential $\omega$ and damping factor $b$ on the membrane potential. F: Log-normal distributions with $\beta^+ = 1.2$ and examplary sample of 80 excitatory (red) and 20 inhibitory (blue) weights. G: A weight with value 0.1 (LIF, blue) and $-0.1$ (LRF, green) was sampled from the log-normal distribution. After rescaling the weight with the normalisation factor $\kappa$ the peak post synaptic potential is equal to the sampled value, independent of intrinsic parameter values.

## 2.2 Error signals

As the dynamics of the teacher and student neuron evolve over time, three possible events can occur when comparing their output spike trains (Figure 1B). First, the student may generate a spike at the same time as the teacher (a *hit*). Second, the teacher may generate a spike when the student does not (a *miss*, denoted by $d^- = -1$). Third, the student may generate a spike when the teacher does not (a *false positive*, denoted by $d^+ = 1$). Since false positives can be detected by the student neuron without feedback, only missed spikes require error signals between teacher and student neurons that are ungraded, unsigned and therefore "unary". Intuitively, a miss occurs when the student's membrane potential $V$ is too low and a false positive occurs when the student's membrane potential is too high. We base learning on this observation and update the student's parameters $\theta$ such that the distance to the threshold is reduced for each miss, but increased for each false positive spike [19, 29].

## 2.3 The event-dependent scaling online learning rule

Our learning rule consists of a gradient-based parameter update like in Memmesheimer et al. [29] (for a derivation see appendix A) and an additional event-dependent scaling factor $\lambda(\Delta d^{\pm})$. We thus call it the Event-Dependent Scaling (EDS) rule. More specifically, we use the partial derivative of the membrane potential with respect to a parameter and perform a gradient ascent step on each miss ($d^- = -1$) and a gradient descent step on each false positive ($d^+ = 1$) spike:

$$\Delta\theta \propto -\lambda(\Delta d^{\pm})d^{\pm}\frac{\partial V(t)}{\partial\theta}. \tag{1}$$

Here, the EDS factor $\lambda(\Delta t)$ scales parameter updates according to the truncated sigmoidal function

$$\lambda(\Delta d^{\pm}) = 1000 - 1000\exp\left(\ln(0.5)\left(\frac{\min(\Delta d^{\pm}, \Delta_{\theta})}{m}\right)^{b}\right), \tag{2}$$

where $\Delta_{\theta}$ is a cutoff parameter, $m$ determines the midpoint and $b$ the slope of the sigmoidal and $\Delta d^{\pm}$ denotes the time since the last parameter update. Thus, the size of the parameter update is event-dependent. In practice, we use $\Delta_{\theta} = 75$, $m = 500$ and $b = 4$, which is plotted in figure 2A. In contrast to the EDS rule, surrogate gradient methods [23, 26, 27] use a scaling function $\lambda(V(t))$ that is based on the current membrane potential. We compare the EDS factor $\lambda(\Delta d^{\pm})$ to the the SuperSpike surrogate gradient $\lambda(V) = (\beta|V - V_{\text{Thr}}| + 1)^{-2}$ [23].

Using the EDS rule, parameter updates are temporal, stochastic and online. Updates are temporal since the underlying function that we optimise is a function evolving over time i.e. the parameterised function that describes the membrane potential. They are stochastic since updates are performed for a single teacher-student pair and not as averages over multiple initialisations. And lastly, they are online since gradients are calculated for the current point in time only, without propagating errors through time [24] and therefore without taking the recurrent nature of neural dynamics into account (cf. [26]). To speed up learning, we use the Adam optimiser [33] with default parameters ($\beta_1 = 0.9$, $\beta_2 = 0.999$, $\epsilon = 1e^{-8}$) and per parameter learning rates, which are scaled according to the range from which target values are sampled ($\eta_w = 35e^{-6}$, $\eta_{\tau_s} = 7e^{-4}$, $\eta_{\tau_m} = 28e^{-4}$, $\eta_{V_r} = 7e^{-5}$ and $\eta_{\tilde{w}} = 8e^{-5}$, $\eta_b = 15e^{-6}$, $\eta_{\omega} = 33e^{-7}$, $\eta_{\tilde{V}_r} = 8e^{-5}$ $\eta_{\tilde{I}_r} = 8e^{-5}$).

## 2.4 Neuron models

Throughout the experiments, we use two neuron models: the leaky integrate-and-fire (LIF) neuron [5] and the leaky resonate-and-fire (LRF) neuron [6] with current-based synaptic inputs. Both of the models are based on linear subthreshold dynamics, a fixed spiking threshold and a post spike reset rule: Whenever the linear dynamics reach the threshold value, the neuron elicits an output spike and its dynamics are reset according to the reset rule. For simplicity, the membrane potential $V$ of both models is rescaled such that the distance from the resting potential ($V_0 = 0$) and the spiking threshold $V_{\text{thr}}$ is 1. Input, output and target spikes are modelled as Dirac delta spike trains $S(t) = \sum_{\{\hat{t}\}}\delta(\Delta\hat{t})$, where $\Delta\hat{t} = t - \hat{t}$ substitutes the time that has elapsed since a spike event $\hat{t}$, and $\{\hat{t}\}$ is a shorthand notation for the set of $n$ spike times of a spike train $\{t_k \mid k = 1, 2, ..., n \text{ and } t_k \leq t\}$. The condition $t_k \leq t$ preserves temporal causality.

Solving the linear system of differential equations that determines the subthreshold voltage dynamics of the leaky **integrate-and-fire** (LIF) neuron and adding the spike reset rule (appendix B.2) yields

the spike-response model [34]. It describes how the membrane potential $V$ of neuron $j$ evolves over time:

$$V_j(t) = \sum_i^N w_{ij} \sum_{\{\hat{t}_i\}} K(\Delta\hat{t}_i) + (V_r - V_{thr}) \sum_{\{\hat{t}_j\}} \exp\left(-\frac{\Delta\hat{t}_j}{\tau_m}\right), \tag{3}$$

with double exponential voltage kernel

$$K(\Delta t) = \exp\left(-\frac{\Delta t}{\tau_m}\right) - \exp\left(-\frac{\Delta t}{\tau_s}\right), \tag{4}$$

input spike times $\hat{t}_i$, output spike times $\hat{t}_j$ and spiking threshold $V_{thr}$. The synaptic weights $w_{ij}$ determine how strongly the membrane potential $V$ is perturbed by synaptic inputs and if the perturbation is excitatory or inhibitory. The synaptic time constant $\tau_s$ and membrane time constant $\tau_m$ determine how quickly synaptic currents and the membrane potential return to rest (Fig. 2C). Together with the reset potential $V_r$, they determine the intrinsic dynamics of the neuron. During training, the intrinsic parameters $(\tau_s, \tau_m, V_r)$ are optimised along with synaptic weights.

The temporal dynamics of the leaky **resonate-and-fire** (LRF) neuron [6] provide a model for phenomena observed in biological neurons that can not be generated by a LIF neuron [1], most prominently sub-threshold oscillations of the membrane potential [35]. This makes LRFs sensitive to inputs of a certain frequency, whereas LIF neurons are sensitive to the rate of inputs. Solving the system of the two coupled linear differential equations (appendix B.3) that describe the oscillating subthreshold potential $\tilde{V}$ and the resonant current $\tilde{I}$ yields

$$\tilde{V}_j(t) = \sum_i^N w_{ij} \sum_{\{\hat{t}_i > \hat{t}_j\}} \exp(\Delta\hat{t}_i b)\sin(\Delta\hat{t}_i\omega) + \exp(\Delta\hat{t}_j b)(\tilde{V}_r\cos(\Delta\hat{t}_j\omega) + \tilde{I}_r\sin(\Delta\hat{t}_j\omega)), \tag{5}$$

with input spike times $\hat{t}_i$ and most recent output spike time $\hat{t}_j$ (Fig. 2D). Akin to a pendulum, the temporal dynamics of the LRF neuron are perturbed by incoming spikes and scaled by the synaptic weights $w_{ij}$. The frequency of the subthreshold oscillations is determined by $\omega \geq 0$, and the dynamics return to rest based on the damping factor $b \leq 0$ (Fig. 2E). In contrast to the LIF neuron, the synaptic currents are reset after each spike, i.e., the dynamics are set to $\tilde{I}_r, \tilde{V}_r$ on the phase plane. Consequentially, in equation 5 we sum only over input spikes since the most recent output spike $\{\hat{t}_i > \hat{t}_j\}$. In summary, the learnable parameters of the LRF neuron are the synaptic weights $w_{ij}$, the damping factor $b$, the frequency of the subthreshold oscillations $\omega$ and the reset values $\tilde{I}_r$ and $\tilde{V}_r$.

## 2.5 Parameter initialisations

**Input spike trains** are homogeneously Poisson distributed. In total, there are 100 synaptic inputs and the ratio between excitatory and inhibitory inputs is $4 : 1$ [36]. Excitatory and inhibitory input firing rates differ accordingly, i.e., $r_{in}^+ = 10$Hz and $r_{in}^- = 40$Hz.

**Synaptic weights** are sampled independently from a log-normal distribution [37]. The mean $\mu_w$ and variance $\sigma_w^2$ of the distribution are set so that approximately 99% of the cumulative distribution function is $\leq 0.2$ and thus below one-fifth of the distance between resting potential and threshold (appendix C). Further, we use the bias term $\beta^+$ to fit the target output firing rate $r_{out}$ by creating a small imbalance between excitatory and inhibitory weights (Fig. 2F):

$$w_{exc} \sim \beta^+\kappa\,\text{Lognormal}(\mu_w, \sigma_w^2) \quad \text{and} \quad w_{inh} \sim \kappa\,\text{Lognormal}(\mu_w, \sigma_w^2). \tag{6}$$

If a weight is $> 0.3$, it is resampled. Weights are then scaled once with the normalisation factor $\kappa$ [38, p. 143], in order for the peak amplitude of the postsynaptic potential to be equivalent to the size of the sampled weight, independent of the values of the intrinsic parameters (Fig. 2G). Note, that $\kappa$ is different for leaky integrate-and-fire and leaky resonate-and-fire neurons (appendix D). To ensure balance between excitation and inhibition, target values are resampled as described in section 2.5 until there is a $\beta^+ < 2.5$ that reaches the target output firing rate $r_{out}$. Importantly, the position of excitatory and inhibitory weights is randomised, i.e. weights do not adhere to Dale's principle [39] during learning and can flip sign.

The **teacher neuron** parameters determine the target computation and are not altered during the training process. Both the student's initial values and the teacher's parameter values are chosen

in the following way: For **Integrate-and-fire neurons**, the membrane time constant is sampled from $\tau_m \sim \mathcal{U}(10\text{ms}, 60\text{ms})$. The synaptic time constant $\tau_s$ is smaller than the membrane time constant and set to $\tau_s = \tau_m/4$. Reset potentials and target firing rates are sampled from $V_r \sim \mathcal{U}(-1.5, 0.9)$ and $r_{\text{out}} \sim \mathcal{U}(1\text{Hz}, 50\text{Hz})$ respectively. For **Resonate-and-fire neurons**, the damping factor $-b \sim \mathcal{U}(20^1/\text{ms}, 120^1/\text{ms})$ and the frequency of the subthreshold oscillations $\omega \sim \mathcal{U}(2\text{Hz}, 25\text{Hz})$ are repeatedly sampled until $\kappa < 4$. The two reset parameters $\tilde{I}_r$ and $\tilde{V}_r$ are both sampled from $\mathcal{U}(-0.8, 0.8)$. Target output firing rates are sampled from $r_{\text{out}} \sim \mathcal{U}(1\text{Hz}, 20\text{Hz})$.

### 2.6 Convergence and robustness

A simulation's **convergence** is determined by the signed relative error between the student's $\theta$ and teacher's $\theta'$ parameter values:

$$\epsilon(\theta, \theta') = \frac{\sum_i (\theta_i - \theta'_i)}{\max(||\theta'||_2, 0.075)}, \tag{7}$$

where 0.075 in the denominator is included to prevent division by zero. The convergence time is the moment the absolute value of the signed relative error $|\epsilon|$ falls below and stays below the parameter dependent threshold values $\Theta_\theta$ for all parameters. For LIF neurons the target threshold values are $\Theta_w = \Theta_{V_r} = 0.15$ and $\Theta_{\tau_s} = \Theta_{\tau_m} = 0.025$ and for LRF neurons $\Theta_{\tilde{w}} = 0.05$, $\Theta_b = \Theta_\omega = 0.025$ and $\Theta_{V_r} = \Theta_{I_r} = 0.1$.

The **robustness** of the learning rule is tested by adding temporal noise $\xi$ to the spike time of the teacher signal. The noise is sampled from a Gaussian distribution $\xi \sim \lfloor \mathcal{N}(\mu = 0, \sigma_\xi^2) \rceil$, where $\lfloor \cdot \rceil$ denotes rounding to the nearest integer and $\sigma_\xi$ is the standard deviation of the noise in ms.

### 2.7 Implementation, simulation and code availability

All simulations are run for $n = 30$ different random seeds in discretised time at a temporal resolution of 1ms. Reported times refer to simulated and not real time, i.e. the time required to run a simulation was several orders of magnitude faster. For example, 30 independent simulations of 12,000 minutes of simulated time at 1ms temporal resolution (as in Fig. 3), take less than 30 minutes of real time on a single AMD Ryzen 5950x. $C$++ code to replicate all simulations and plots is publicly available[1] under *GPLv3* license and uses the *MPL 2.0* licensed Eigen software library v3.3.7 [40].

## 3 Results

The results are separated into a theoretical and an empirical part. In the theoretical part, we derive the partial derivatives (sec. 3.1) that are required for the parameter update rule (eq. 1) and motivate their event-based versions (sec. 3.2 and derivation in appendix E). In the empirical part, we first show that the EDS rule can recover teacher's parameters in LIF and LRF neurons (sec. 3.3). Subsequently, we investigate the influence and functionality of the EDS factor and compare it to the SuperSpike surrogate gradient [23] in a lesion study (sec. 3.4). Finally, we test the robustness of the algorithm to temporal noise in the teacher signal (sec. 3.5).

### 3.1 Gradients

**Integrate-and-fire neuron:** Without explicit derivation, the partial derivatives of the membrane potential $V$ (eq. 3) with respect to one of the synaptic weights $w_{ij}$, the synaptic time constant $\tau_s$, membrane time constant $\tau_m$ and the reset potential $V_r$ are

$$\frac{\partial V_j(t)}{\partial w_{ij}} = \sum_{\{\hat{t}_i\}} K(\Delta \hat{t}_i), \tag{8}$$

$$\frac{\partial V_j(t)}{\partial \tau_s} = -\frac{1}{\tau_s^2} \sum_i^N w_{ij} \sum_{\{\hat{t}_i\}} \Delta \hat{t}_i \exp\left(-\frac{\Delta \hat{t}_i}{\tau_s}\right), \tag{9}$$

---

[1]https://github.com/lukas-braun/learning-neural-computations

$$\frac{\partial V_j(t)}{\partial \tau_m} = \frac{1}{\tau_m^2} \sum_i^N w_{ij} \sum_{\{\hat{t}_i\}} \Delta \hat{t}_i \exp\left(-\frac{\Delta \hat{t}_i}{\tau_m}\right) + \frac{1}{\tau_m^2}(V_r - V_{thr}) \sum_{\{\hat{t}_j\}} \Delta \hat{t}_j \exp\left(-\frac{\Delta \hat{t}_j}{\tau_m}\right), \quad (10)$$

$$\frac{\partial V_j(t)}{\partial V_r} = \sum_{\{\hat{t}_j\}} \exp\left(-\frac{\Delta \hat{t}_j}{\tau_m}\right). \quad (11)$$

**Resonate-and-fire neuron:** Similarly, deriving the partial derivatives for the membrane potential $\tilde{V}$ (eq. 5) of the leaky resonate-and-fire neuron with respect to a single synaptic weight $w_{ij}$, the damping factor $b$, the frequency of the subthreshold oscillations $\omega$ and the two reset values $\tilde{V}_r$ and $\tilde{I}_r$ results in

$$\frac{\partial \tilde{V}_j(t)}{\partial w_{ij}} = \sum_{\{\hat{t}_i > \hat{t}_j\}} \exp(\Delta \hat{t}_i b) \sin(\Delta \hat{t}_i \omega), \quad (12)$$

$$\frac{\partial \tilde{V}_j(t)}{\partial b} = \sum_i^N w_{ij} \sum_{\{\hat{t}_i > \hat{t}_j\}} \Delta \hat{t}_i e(\Delta \hat{t}_i b) \sin(\Delta \hat{t}_i \omega) + \Delta \hat{t}_j e(\Delta \hat{t}_j b)(\tilde{V}_r \cos(\Delta \hat{t}_j \omega) + \tilde{I}_r \sin(\Delta \hat{t}_j \omega)),$$
$$(13)$$

$$\frac{\partial \tilde{V}_j(t)}{\partial \omega} = \sum_i^N w_{ij} \sum_{\{\hat{t}_i > \hat{t}_j\}} \Delta \hat{t}_i e(\Delta \hat{t}_i b) \cos(\Delta \hat{t}_i \omega) + \Delta \hat{t}_j e(\Delta \hat{t}_j b)(\tilde{I}_r \cos(\Delta \hat{t}_j \omega) - \tilde{V}_r \sin(\Delta \hat{t}_j \omega)),$$
$$(14)$$

$$\frac{\partial \tilde{V}_j(t)}{\partial \tilde{V}_r} = \exp(\Delta \hat{t}_j b) \sin(\Delta \hat{t}_j \omega) \quad (15) \qquad \text{and} \qquad \frac{\partial \tilde{V}_j(t)}{\partial \tilde{I}_r} = \exp(\Delta \hat{t}_j b) \cos(\Delta \hat{t}_j \omega). \quad (16)$$

### 3.2 Event-based dynamics

The closed form solution of the membrane potential dynamics and its partial derivatives comprise sums over input and output spike times. These are computationally expensive and create a large memory footprint, because all past spike times have to be stored and summed over. Since the summands are decaying exponential functions, one approximate solution is to only sum over terms that are still contributing significantly. Here, instead of relying on an approximation, we show that one can replace these sums by factorising the exponentials. As a result, the sums over spike times are replaced by a recurrence relation, which depends on constants that are updated on each spike event only. Using these event-based dynamics, both the computational as well as the memory complexity can be significantly reduced. For all factorised equations and their derivations see appendix E.

### 3.3 Joint online learning of synaptic weights and intrinsic parameters

We asked whether it is possible to learn a neural computation from sparse temporal feedback. We found that, indeed, the EDS rule can be used to recover teacher's parameters in LIF and LRF neurons for biologically plausible initial and target parameter values (Fig. 3A-B). The learning dynamics contract but do not converge to the solution i.e., parameter updates fluctuate around the solution without further decay (appendix F). As a consequence, the size of the fluctuations and therefore the final precision of the solution is dependent on the learning rate. Since learning happens online, the student can also recover teachers that vary over time (appendix G).

During training, teacher and student output spike trains become more correlated (Fig. 3C,E). After training, an average of $94\% \pm 3\%$ SE of LIF and $96.3\% \pm 3\%$ SE of LRF output spike times match the teachers' output exactly. For LIF neurons, $3\%$ of the remaining spikes fall within 1ms before, and $2.3\%$ within 1ms after the target time. For LRF neurons, these values are $1.4\%$ and $1.9\%$. Which, combined, addresses more than $99\%$ of all spikes. Final paramter fluctuations in LIF neurons, especially in the synaptic weights, are higher than in LRF neurons, possibly because synaptic currents

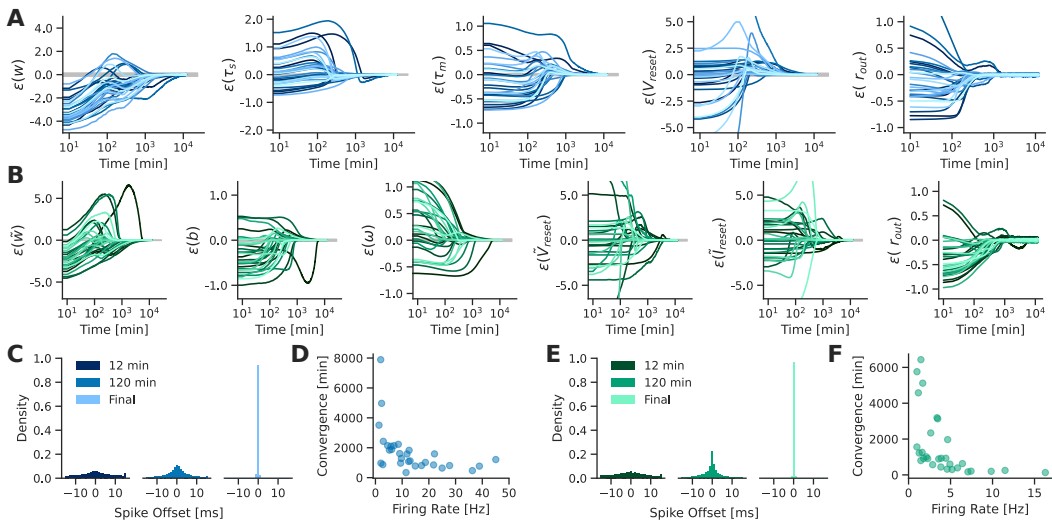

Figure 3: Learning dynamics. A, B: Signed relative error (eq. 7) between the students' and teachers' parameters over time. Parameter trajectories converge to and stay within the error margins (grey). A: LIF neuron parameters, from left to right, are: synaptic weights, synaptic time constant, membrane time constant, reset potential and target firing rate and accordingly for B: LRF neurons, synaptic weights, damping factor, frequency of the subthreshold oscillations, reset potential, reset current and target firing rate. C, E: Probability distribution of students' output spike times around target spikes for LIF (C) and LRF (E) neurons after $0.1\%$, $1\%$ and $100\%$ training time. D, F: Change of convergence time with respect to target firing rates for LIF (D) and LRF (F) neurons.

are not reset after output spikes in LIF neurons, making credit assignment more difficult. For both neuron models, the convergence times decrease as the target firing rates increase (Fig. 3D,F). This is expected, since parameter updates occur after each missed or false positive spike and therefore higher firing rates result in more parameter updates in the same amount of training time.

Importantly, synaptic weights and output firing rates deviate from their target values and performance is significantly inclined when restricting training to synaptic weights. For LIF (and LRF) neurons only $18.5\% \pm 0.7\%$ SE (and $12.9\% \pm 0.6\%$ SE) of output spikes are direct hits and an additional $25.3\%$ (and $19.8\%$) fall within the 1ms window around the target spike, addressing for less than $44\%$ (and $33\%$) of total spikes (appendix H).

### 3.4 Influence and necessity of the event-dependent scaling factor

If we use the "vanilla" parameter update rule (eq. 1 without $\lambda(\Delta d^{\pm})$), synaptic weights vanish, and the intrinsic parameters diverge or settle at a constant but wrong value, i.e., the training fails (appendix I). We designed the EDS factor such that parameter updates are not immediately reversed by the next error signal, i.e., consecutive parameter updates are down-scaled as a function of their temporal proximity. To better understand the effect of the EDS factor, we trained parameters separately, as well as in all possible combinations. To this end, we optimised subsets of the student's parameters and kept all other parameters fixed to their optimal, target values. In LIF neurons, single parameters, except the reset potential, can be learned reliably without the EDS factor, whereas optimisation of parameter combinations fail (Fig. 4A). Surprisingly, the performance of the SuperSpike surrogate is equal to or worse than the vanilla learning rule (except for the pair $w, \tau_m$). This is due to local minima, or degenerate solutions in which the firing rate is kept stable while parameters diverge (appendix K), rather than larger than permissible noise levels (section 2.6). When optimising synaptic weights only, the recovery rate increases as, through changing $\beta$, the surrogate is getting more similar to the vanilla learning rule (Fig. 4B). A similar trend can be observed for LRF neurons, where the performance of surrogate gradients falls below the vanilla performance (Fig. 4C,D plot with all parameter combinations in appendix J) and the performance of surrogate gradients increases the more similar it becomes to and settles at the optimal performance of vanilla learning. Our learning rule performs well for LRF neurons across all parameter combinations. In fact, including more parameters is helping in situation where a single parameter would get stuck in a local minima (data not shown).

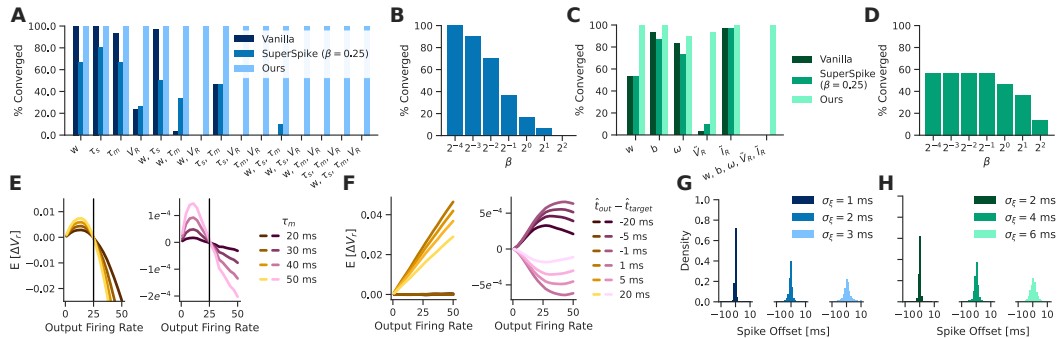

Figure 4: Event-dependent scaling factor and temporal noise. A, C: Percentage of $n = 30$ students that converged to the target values for all possible parameter combinations with vanilla, surrogate gradient and EDS algorithm in LIF (A) and LRF (C) neurons. B, D: Training only synaptic weights with surrogate gradients for varying $\beta$ values in LIF (B) and LRF (D) neurons. E, F: Expected gradient value of the partial derivative with respect to $V_r$ for uncorrelated (E) and correlated (F) Poisson spike trains without (left) and with EDS factor (right). G, H: Probability distribution of students' output spike times around target spikes for three noise levels in the target spike times.

Further insights into why the EDS factor is so effective, can be acquired by analysing expected parameter updates. For example, if the target and output spike trains of a LIF neuron are uncorrelated Poisson spike trains, the expected vanilla parameter updates with respect to the reset potential $E[\Delta V_r]$ are accurately positive if the output firing rate is too low but negative if the output firing rate is too high (Fig. 4E). However, as training progresses, output and target spike trains become highly correlated. Accordingly, if we calculate $E[\Delta V_r]$ with the vanilla learning rule for an output-spike train that is a shifted version of the target spike train, i.e. highly correlated, the average parameter updates are distorted (Fig. 4F, left). They linearly increase for positive offsets but stay around zero for negative offsets, leading to inaccurate parameter updates. In contrast, the EDS rule disentangles $E[\Delta V_r]$, such that too early spikes lead to positive updates and too late spikes to negative updates of equal size, accurately leading to a respectively reduced or increased reset potential (Fig. 4F, right).

### 3.5 Influence of temporal noise in the teacher signal

Adding temporal, Gaussian noise with standard deviation $\sigma_\xi$ to the target spike times decreases the precision of the solution found by the student as noise levels increase, but does generally not brake the algorithm (Fig. 4G,H). However, LRF neurons with target firing rates above 35Hz start to diverge at the highest noise level (data not shown). This is not surprising, when we consider that the average inter-spike-interval at 35Hz is approximately 29ms and temporal noise at $\sigma_\xi = 3$ frequently yields perturbations $> 5$ms, rendering noise indistinguishable from signal. Considering the lower average target firing rates in LRF neurons in comparison to LIF neurons, also explains why higher noise levels yield to, on average, similar precision in LRF comparing to LIF neurons at varying noise levels. More generally, adding temporal noise to the feedback signal reveals a subtle relation between firing rates, noise variance and the resulting permissible signal to noise ratio.

## 4 Discussion

Learning of input-output functions from a continuous stream of information is difficult. We proposed a gradient-based update rule capable of learning the unique parameters that determine a neural computation from ungraded, unsigned and temporally sparse feedback signals. Importantly, EDS learns online and without backpropagating errors backward in time, i.e., without explicit memory traces of input or target spike times, or past states of the neuron. Instead, we derived an efficient and exact implementation of membrane potential dynamics and their gradients. In contrast to previous theoretical work, but in alignment with observations in biological neurons [3, 4], we regard intrinsic parameters like time constants etc. as crucial components of learning. Therefore, in line with recent theoretical work suggesting that networks benefit from heterogeneous neural computations [41, 42], we optimise the intrinsic parameters along the synaptic weights, and further, derive the learning rule for two neuron models that implement different computations: the frequently used and optimised LIF

neuron and the LRF neuron. Whether EDS can be extended to non-linear models like quadratic [43] or exponential LIF [44] or Izhikevich [1] neurons must be investigated.

Numerical simulations show that our algorithm can learn exact neural computations across a wide range of biologically plausible target values. The EDS rule can adapt to changing target computations mid-flight and is robust against temporal noise in the feedback signal. Optimising different combinations of parameters revealed the crucial role played by the EDS factor $\lambda(\Delta d^{\pm})$. Surprisingly, the EDS factor cannot be replaced by a surrogate gradient. While surrogate gradients are the current state of the art to train spiking multi-layer neural networks with backpropagation through time, they may not be optimal to learn neural computations that go beyond rate coding. More specifically, the observation that surrogate gradients produce diverging parameter values but stabilised firing rates suggests that mutual online learning of intrinsic parameters and synaptic weights from target spike times requires a time-dependent scaling factor in contrast to the voltage-dependent scaling of surrogate gradients.

It would be interesting, but brake the online learning assumption, to replace surrogate gradients with EDS factors in multi-layer networks and train them with backpropagation through time. An online learning alternative could use EDS to extend difference target propagation [45, 46] from rate-coded neurons to spiking neural networks. In such a network, the parameters of neurons that provide the downstream signals are optimised to predict spike times of neurons in lower layers, and parameters of upstream neurons are updated such that the backpropagated target spike times are generated.

## 4.1 Limitations and conclusions

While we motivate the EDS factor $\lambda(\Delta d^{\pm})$ by means of its empirical success, a rigorous theoretical explanation for its function is missing. An analytical derivation of the expected gradient values for correlated output and target spike trains in spirit of figure 4 E and F could further unveil the underlying principles of our algorithm.

In LRF neurons target computations with sustained subthreshold oscillations ($-b \ll 20^{1}/\mathrm{ms}$) or spike bursts, i.e., dynamics that reset outside of the separatrix, cannot be learned reliably (data not shown). In addition, target firing rates in LRF neurons remain low, due to the difficulty of achieving high output firing rates from uncorrelated Poisson inputs while keeping a balance between excitation and inhibition. If our algorithm also works for correlated, oscillating or bursting inputs near the neuron's eigenfrequency, which cause high output firing rates, is left for future studies.

Further, convergence to target values is slow (up to $\approx 5.5$ and $\approx 4.3$ days of simulated time in LIF and LRF neurons respectively). As a proof of principle, we started each simulation at a random point in parameter space and required convergence to a second, randomly sampled point in parameter space. This could lead to long learning trajectories and slow convergence times (see also [47]), especially when target firing rates were low. In contrast, the parameter subspace that results in good performance in supervised rate-coded networks is large [48], and spiking neural networks may solve a task with a wide range of different architectures or parameters [27]. From a biological perspective, most electrophysiological changes induced by plasticity mechanisms are gradual and a complete reorganisation of a neuron's electrophysiology (as in our study) implausible. Ultimately, training speed could be increased dramatically (>10x) by increasing learning rates at the cost of losing precision (appendix F) or by using a learning rate schedule to speed up initial learning and subsequently gain precision as the learning rate decays. Convergence time could be further reduced by batch-learning and when modelling biological data, by using educated guesses for initial model parameters (e.g. pick parameters such that the initial firing rate of the model and data are identical).

We conclude that the complex temporal dynamics of neural computations can be learned online with a local gradient-based update rule from feedback signals that indicate target spike times. Our event-dependent scaling rule will help to bring the success of gradient-based optimisation to spiking neural networks performing heterogeneous computations and contribute to the exploration of gradient-based learning in the brain.

## Acknowledgments and Disclosure of Funding

We would like to thank Professor Dr. Henning Sprekeler for his valuable suggestions and Dr. Andrew Saxe, Milan Klöwer and Anna Wallis for their constructive feedback on the manuscript. Lukas Braun was supported by the Network of European Neuroscience Schools through their NENS Exchange Grant program, by the European Union through their European Community Action Scheme for the Mobility of University Students, the Woodward Scholarship awarded by Wadham College, Oxford and the Medical Research Council [MR/N013468/1]. Tim P. Vogels was supported by a Wellcome Trust Senior Research Fellowship [214316/Z/18/Z].

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
