# A  Derivation of the parameter update rule

The parameter update rule in equation 1 can be derived from the linear distance loss [19, 29] between the membrane potential and the spiking threshold

$$\mathcal{L}(V(t), V_{thr}) = V(t) - V_{thr},$$  (17)

by first taking the partial derivative of the loss function

$$\frac{\partial L(V(t), V_{\text{thr}})}{\partial \theta} = \frac{\partial}{\partial \theta} V(t) - \frac{\partial}{\partial \theta} V_{\text{thr}} = \frac{\partial V(t)}{\partial \theta}$$  (18)

and then setting the direction of the gradient depending on the error signal. Error signals are generated by comparing the output spike trains of the teacher and student neurons. When the student generates a false positive spike, the distance to the threshold has to be maximised and when the student misses a target spike, the distance to the threshold has to be minimised, leading to $d^+ = 1$ and $d^- = -1$ respectively and the gradient

$$\nabla \mathcal{L}(t) = d^{\pm} \frac{\partial V(t)}{\partial \theta}.$$  (19)

Inserting the gradient into the formula for gradient descent optimisation

$$\theta_t = \theta_{t-1} - \eta \nabla \mathcal{L},$$  (20)

with learning rate $\eta$, leads to

$$\Delta \theta \propto -d^{\pm} \frac{\partial V(t)}{\partial \theta},$$  (21)

which is the parameter update rule.

# B  Derivation of the spike-response models

## B.1  Linear systems

Linear systems of first order differential equations [49, p. 349-365] of the form

$$\bar{y}' = A\bar{y} + q(t)$$  (22)

are solved by

$$\bar{y}(t) = \exp(At)\bar{y}(0) + \int_0^t \exp(A(t-s))\bar{q}(s)\mathrm{d}s.$$  (23)

If $A$ has $n$ independent eigenvectors, we can find $\exp(At)$ by using the identity

$$\exp(At) = V \exp(\Lambda t) V^{-1},$$  (24)

where $V = [\bar{v}_1, \bar{v}_2, ..., \bar{v}_n]$ is the eigenvector matrix, $V^{-1}$ is the inverse of the eigenvector matrix and $\Lambda = \mathrm{diag}(\lambda_1, \lambda_2, ..., \lambda_n)$ is the eigenvalue matrix.

## B.2  Dynamics of the leaky integrate-and-fire neuron

The dynamics of the leaky integrate-and-fire neuron with current-based synaptic inputs are determined by a linear system of two differential equations and a voltage dependent reset rule, which resets the membrane potential $V(t)$ back to $V_\mathrm{r}$ if it hits the threshold potential $V(t) = V_\mathrm{thr}$. By contrast, the input current $I(t)$ evolves continuously and is never reset. The membrane potential $V$ and the synaptic current $I$ of the $j$-th postsynaptic neuron in the network evolve according to

$$\tau_m \frac{d}{dt} V_j = -V + V_0 + R_m I_j(t) \quad (25) \qquad \text{and} \qquad \frac{d}{dt} I_j = -\frac{I_j}{\tau_s} + \sum_i^N w_{ij} S_i(t), \quad (26)$$

with resting potential $V_0$, synaptic time constant $\tau_s$, membrane time constant $\tau_m = R_m C_m$, membrane resistance $R_m$, membrane capacitance $C_m$, synaptic weights $w_{ij}$, and input spike trains

$S_i(t) = \sum_{\{\hat{t}_i\}} \delta(\Delta \hat{t}_i)$. Rewriting the equations as a first order system of linear differential equations yields

$$\underbrace{\frac{d}{dt}\begin{bmatrix} I_j \\ V_j \end{bmatrix}}_{\bar{y}'} = \underbrace{\begin{bmatrix} -\frac{1}{\tau_s} & 0 \\ R_m/\tau_m & -\frac{1}{\tau_m} \end{bmatrix}}_{A} \underbrace{\begin{bmatrix} I_j \\ V_j \end{bmatrix}}_{\bar{y}} + \underbrace{\begin{bmatrix} \sum_i^N w_{ij} \sum_{\{\hat{t}_i\}} \delta(\Delta \hat{t}_i) \\ V_0/\tau_m \end{bmatrix}}_{q(t)}. \tag{27}$$

In order to solve the equation, we first have to find the eigenvectors and eigenvalues of $A$.

## B.2.1 Eigenvalues

Finding the characteristic equation

$$\begin{aligned}
\det &\begin{pmatrix} -\frac{1}{\tau_s} - \lambda & 0 \\ R_m/\tau_m & -\frac{1}{\tau_m} - \lambda \end{pmatrix} \\
&= \left( -\frac{1}{\tau_s} - \lambda \right)\left( -\frac{1}{\tau_m} - \lambda \right) \\
&= \lambda^2 + \frac{1}{\tau_s}\lambda + \frac{1}{\tau_m}\lambda + \frac{1}{\tau_s \tau_m} \\
&= \lambda^2 + \frac{\tau_s + \tau_m}{\tau_s \tau_m}\lambda + \frac{1}{\tau_s \tau_m}
\end{aligned} \tag{28}$$

and solving it

$$\begin{aligned}
&-\frac{\tau_s + \tau_m}{2\tau_s \tau_m} \pm \sqrt{\left( \frac{\tau_s + \tau_m}{2\tau_s \tau_m} \right)^2 - \frac{1}{\tau_s \tau_m}} \overset{!}{=} 0 \\
\Leftrightarrow \quad &\frac{-\tau_s - \tau_m}{2\tau_s \tau_m} \pm \sqrt{\frac{\tau_s^2 + 2\tau_s \tau_m + \tau_m^2}{4\tau_s \tau_m} - \frac{4\tau_s \tau_m}{4\tau_s^2 \tau_m^2}} = 0 \\
&\Leftrightarrow \quad \frac{-\tau_s - \tau_m}{2\tau_s \tau_m} \pm \frac{\tau_s - \tau_m}{2\tau_s \tau_m} = 0
\end{aligned} \tag{29}$$

leads to the eigenvalue matrix

$$\Lambda = \begin{bmatrix} -1/\tau_s & 0 \\ 0 & -1/\tau_m \end{bmatrix}. \tag{30}$$

## B.2.2 Eigenvectors

Next, we find the two corresponding eigenvectors $\bar{v}_1$ and $\bar{v}_2$ by solving $(A - \lambda_{1|2}I)\bar{x} = \bar{0}$. For $\bar{v}_1$ that is

$$\begin{aligned}
&\left( \begin{bmatrix} -1/\tau_s & 0 \\ R_m/\tau_m & -1/\tau_m \end{bmatrix} - \begin{bmatrix} -1/\tau_s & 0 \\ 0 & -1/\tau_s \end{bmatrix} \right) \begin{bmatrix} x_1 \\ x_2 \end{bmatrix} = \begin{bmatrix} 0 \\ 0 \end{bmatrix} \\
&\Leftrightarrow \begin{bmatrix} 0 & 0 \\ R_m/\tau_m & -1/\tau_m + 1/\tau_s \end{bmatrix} \begin{bmatrix} x_1 \\ x_2 \end{bmatrix} = \begin{bmatrix} 0 \\ 0 \end{bmatrix}
\end{aligned} \tag{31}$$

with equation $0x_1 + 0x_2 = 0$ revealing that we can choose $x_2$ freely ($x_2 = c$), which then leads to the second equation

$$\begin{aligned}
\frac{R_m}{\tau_m}x_1 + \left( -\frac{1}{\tau_m} + \frac{1}{\tau_s} \right)c &= 0 \\
\Leftrightarrow x_1 &= c\left( \frac{\tau_s - \tau_m}{R_m \tau_s} \right)
\end{aligned} \tag{32}$$

and therefore to

$$\bar{v}_1 = c\begin{bmatrix} (\tau_s - \tau_m)/R_m \tau_s \\ 1 \end{bmatrix}. \tag{33}$$

Analogically we find $\bar{v}_2$

$$\begin{aligned}
&\left( \begin{bmatrix} -1/\tau_s & 0 \\ R_m/\tau_m & -1/\tau_m \end{bmatrix} - \begin{bmatrix} -1/\tau_m & 0 \\ 0 & -1/\tau_m \end{bmatrix} \right) \begin{bmatrix} x_1 \\ x_2 \end{bmatrix} = \begin{bmatrix} 0 \\ 0 \end{bmatrix} \\
&\Leftrightarrow \begin{bmatrix} -1/\tau_s - 1/\tau_m & 0 \\ R_m/\tau_m & 0 \end{bmatrix} \begin{bmatrix} x_1 \\ x_2 \end{bmatrix} = \begin{bmatrix} 0 \\ 0 \end{bmatrix},
\end{aligned} \tag{34}$$

which leads to a free choice of $x_2 = c$ and $x_1 = 0$ and therefore to the second eigenvector

$$\bar{v}_2 = c \begin{bmatrix} 0 \\ 1 \end{bmatrix}. \tag{35}$$

The complete eigenvector matrix is then

$$V = \begin{bmatrix} (\tau_s - \tau_m)/R_m\tau_s & 0 \\ 1 & 1 \end{bmatrix}. \tag{36}$$

### B.2.3  Inverse of eigenvector matrix

Finally, we have to find the inverse of the eigenvector matrix:

$$
\begin{aligned}
[VI] &= \begin{bmatrix} (\tau_s - \tau_m)/R_m\tau_s & 0 & 1 & 0 \\ 1 & & 1 & 0 & 1 \end{bmatrix} \cdot R_m\tau_s/(\tau_s - \tau_m) \\
&= \begin{bmatrix} 1 & 0 & R_m\tau_s/(\tau_s - \tau_m) & 0 \\ 1 & 1 & 0 & 1 \end{bmatrix} \text{II - I} \\
&= \begin{bmatrix} 1 & 0 & R_m\tau_s/(\tau_s - \tau_m) & 0 \\ 0 & 1 & -R_m\tau_s/(\tau_s - \tau_m) & 1 \end{bmatrix} = [IV^{-1}].
\end{aligned}
\tag{37}
$$

### B.2.4  Solution

Using the identity in equation 24, we can then solve the matrix exponential:

$$
\begin{aligned}
\exp(At) &= V \exp(\Lambda t) V^{-1} \\
&= \begin{bmatrix} \frac{\tau_s - \tau_m}{R_m\tau_s} & 0 \\ 1 & 1 \end{bmatrix} \begin{bmatrix} \exp\left(\frac{-t}{\tau_s}\right) & 0 \\ 0 & \exp\left(\frac{-t}{\tau_m}\right) \end{bmatrix} V^{-1} \\
&= \begin{bmatrix} \frac{\tau_s - \tau_m}{R_m\tau_s} \exp\left(\frac{-t}{\tau_s}\right) & 0 \\ \exp\left(\frac{-t}{\tau_s}\right) & \exp\left(\frac{-t}{\tau_m}\right) \end{bmatrix} \begin{bmatrix} \frac{R_m\tau_s}{\tau_s - \tau_m} & 0 \\ \frac{-R_m\tau_s}{\tau_s - \tau_m} & 1 \end{bmatrix} \\
&= \begin{bmatrix} \exp\left(\frac{-t}{\tau_s}\right) & 0 \\ -\frac{R_m\tau_s}{\tau_s - \tau_m}\left(\exp\left(\frac{-t}{\tau_m}\right) - \exp\left(\frac{-t}{\tau_s}\right)\right) & \exp\left(\frac{-t}{\tau_m}\right) \end{bmatrix}.
\end{aligned}
\tag{38}
$$

Note that $-\frac{R_m\tau_s}{\tau_s - \tau_m} > 0$ since $0 < \tau_s < \tau_m$ and therefore the sign of the weights in later substitution steps is preserved. In the following we substitute

$$K(t) = \exp\left(-\frac{t}{\tau_m}\right) - \exp\left(-\frac{t}{\tau_s}\right) \tag{39}$$

and call $K(t)$ the voltage kernel. Using the solution of the matrix exponential, we can then derive the null solution

$$
\begin{aligned}
V_n(t) &= \exp(At)\bar{y}(0) \\
&= \begin{bmatrix} \exp\left(\frac{-t}{\tau_s}\right) & 0 \\ -\frac{R_m\tau_s}{\tau_s - \tau_m}K(t) & \exp\left(\frac{-t}{\tau_m}\right) \end{bmatrix} \bar{y}(0) \\
&= \begin{bmatrix} c_1 \exp\left(\frac{-t}{\tau_s}\right) \\ -c_1\frac{R_m\tau_s}{\tau_s - \tau_m}K(t) + c_2 \exp\left(\frac{-t}{\tau_m}\right) \end{bmatrix}
\end{aligned}
\tag{40}
$$

and the particular solution

$$V_p(t) = \int_0^t \exp(A(t-s))\bar{q}(s)\mathrm{d}s$$

$$= \int_0^t \begin{bmatrix} \exp\left(\frac{-(t-s)}{\tau_s}\right) & 0 \\ -\frac{R_m\tau_s}{\tau_s-\tau_m}K(t-s) & \exp\left(\frac{-(t-s)}{\tau_m}\right) \end{bmatrix} \begin{bmatrix} \sum_i^N w_{ij}\sum_{\{\hat{t}_i\}}\delta(s-t_i) \\ V_0/\tau_m \end{bmatrix} \mathrm{d}s$$

$$= \begin{bmatrix} \sum_i^N w_{ij}\sum_{\{\hat{t}_i\}}\int_0^t \exp\left(\frac{-(t-s)}{\tau_s}\right)\delta(s-t_i)\mathrm{d}s \\ -\frac{R_m\tau_s}{\tau_s-\tau_m}\sum_i^N w_{ij}\sum_{\{\hat{t}_i\}}\int_0^t K(t-s)\delta(s-t_i)\mathrm{d}s \end{bmatrix} + \begin{bmatrix} 0 \\ V_0/\tau_m \int_0^t \exp\left(\frac{-(t-s)}{\tau_m}\right)\mathrm{d}s \end{bmatrix} \tag{41}$$

$$= \begin{bmatrix} \sum_i^N w_{ij}\sum_{\{\hat{t}_i\}}\exp\left(\frac{t_i-t}{\tau_s}\right) \\ -\frac{R_m\tau_s}{\tau_s-\tau_m}\sum_i^N w_{ij}\sum_{\{\hat{t}_i\}}K(t-\hat{t}_i) \end{bmatrix} + \begin{bmatrix} 0 \\ V_0 - V_0\exp\left(\frac{-t}{\tau_m}\right) \end{bmatrix}.$$

The final solution is then the sum of the null and particular solution, which yields

$$I_j(t) = -c_1\exp\left(\frac{-t}{\tau_s}\right) + \sum_i^N w_{ij}\sum_{\{\hat{t}_i\}}\exp\left(\frac{t_i-t}{\tau_s}\right) \tag{42}$$

for the input current and

$$V_j(t) = -c_1\frac{R_m\tau_s}{\tau_s-\tau_m}K(t) + c_2\exp\left(\frac{-t}{\tau_m}\right) - \frac{R_m\tau_s}{\tau_s-\tau_m}\sum_i^N w_{ij}\sum_{\{\hat{t}_i\}}K(t-\hat{t}_i) + V_0 - V_0\exp\left(\frac{-t}{\tau_m}\right) \tag{43}$$

for the membrane potential.

By setting the initial conditions for the input current and the membrane potential to $c_1 = 0$ and $c_2 = V_0$ respectively and by including the factor $-R_m\tau_s/\tau_s - \tau_m$ in the synaptic weights we arrive at the spike response model [34]

$$V_j(t) = V_0 + \sum_i^N w_{ij}\sum_{\{\hat{t}_i\}}K(t-\hat{t}_i). \tag{44}$$

This equation describes only the dynamics of the membrane potential below the threshold. By adding a voltage reset at output spike times $\hat{t}_j$, i.e. when the membrane potential reaches the threshold $V(t) = V_{\text{thr}}$, we arrive at the final formula for the dynamics of the leaky resonate-and-fire neuron

$$V_j(t) = V_0 + \sum_i^N w_{ij}\sum_{\{\hat{t}_i\}}K(t-\hat{t}_i) + (V_r - V_{\text{thr}})\sum_{\{\hat{t}_j\}}\exp\left(\frac{-(t-\hat{t}_j)}{\tau_m}\right). \tag{45}$$

## B.3 Dynamics of the leaky resonate-and-fire neuron

In this section, we derive the dynamics of the leaky resonate-and-fire neuron. They are determined by a voltage dependent reset rule and by two coupled linear differential equations for the synaptic currents $\tilde{I}$ and membrane potential $\tilde{V}$:

$$\frac{d}{dt}\tilde{V}_j = \omega\tilde{I} + b\tilde{V}_j \qquad (46) \qquad \text{and} \qquad \frac{d}{dt}\tilde{I}_j = b\tilde{I} - \omega\tilde{V}_j + \sum_i^N w_{ij}\tilde{S}_i(t), \quad (47)$$

with damping factor $b$, frequency of the subthreshold oscillations $\omega$, synaptic weights $w_{ij}$ and input spike trains $\tilde{S}_i(t) = \sum_{\{\hat{t}_i > \hat{t}_j\}}\delta(\Delta\hat{t}_i)$. Rewriting the equations as a linear system yields

$$\underbrace{\frac{d}{dt}\begin{bmatrix}\tilde{I}_j \\ \tilde{V}_j\end{bmatrix}}_{\bar{y}'} = \underbrace{\begin{bmatrix} b & -\omega \\ \omega & b \end{bmatrix}}_{A}\underbrace{\begin{bmatrix}\tilde{I}_j \\ \tilde{V}_j\end{bmatrix}}_{\bar{y}} + \underbrace{\begin{bmatrix}\sum_i^N w_{ij}\sum_{\{\hat{t}_i\}}\delta(\Delta\hat{t}_i) \\ 0\end{bmatrix}}_{q(t)}. \tag{48}$$

If if the membrane potential $\tilde{V}(t)$ hits the threshold $\tilde{V}_{\text{thr}} = 1$, both the membrane potential and the resonant current are reset to $\tilde{V}_r$ and $\tilde{I}_r$ respectively. This is in contrast to the leaky integrate-and-fire neuron where only the membrane potential but not the synaptic currents are reset. Following section B.1, we next derive the eigenvalues and eigenvectors of the linear system.

### B.3.1 Eigenvalues

Finding the Eigenvalues first requires finding the characteristic equation

$$\det \begin{pmatrix} b - \lambda & -\omega \\ \omega & b - \lambda \end{pmatrix}$$
$$= (b - \lambda)(b - \lambda) - (-\omega\omega) \tag{49}$$
$$= \lambda^2 - 2b\lambda + b^2 + \omega^2$$

and then solving it

$$b \pm \sqrt{b^2 - (b^2 + \omega^2)} \stackrel{!}{=} 0$$
$$\Leftrightarrow b \pm i\omega, \tag{50}$$

leading to the corresponding eigenvalue matrix

$$\Lambda = \begin{bmatrix} b + i\omega & 0 \\ 0 & b - i\omega \end{bmatrix}. \tag{51}$$

### B.3.2 Eigenvectors

Next, we find the eigenvectors $\bar{v}_1$ and $\bar{v}_2$ by solving $(A - \lambda_{1|2}I)\bar{x} = \bar{0}$

$$\left( \begin{bmatrix} b & -\omega \\ \omega & b \end{bmatrix} - \begin{bmatrix} b \pm i\omega & 0 \\ 0 & b \pm i\omega \end{bmatrix} \right) \begin{bmatrix} x_1 \\ x_2 \end{bmatrix} = \begin{bmatrix} 0 \\ 0 \end{bmatrix}. \tag{52}$$

For the first eigenvalue, this is

$$\begin{bmatrix} -i\omega & -\omega \\ \omega & -i\omega \end{bmatrix} \begin{bmatrix} x_1 \\ x_2 \end{bmatrix} = \begin{bmatrix} 0 \\ 0 \end{bmatrix} \quad \text{II} - i \cdot \text{I}$$
$$\Leftrightarrow \begin{bmatrix} -i\omega & -\omega \\ 0 & 0 \end{bmatrix} \begin{bmatrix} x_1 \\ x_2 \end{bmatrix} = \begin{bmatrix} 0 \\ 0 \end{bmatrix}, \tag{53}$$

which leads to $x_2 = c$ and

$$-i\omega x_1 - c\omega = 0$$
$$\Leftrightarrow x_1 = -\frac{c\omega}{i\omega} \tag{54}$$
$$\Leftrightarrow x_1 = ci$$

and therefore to

$$barv_1 = c \begin{bmatrix} i \\ 1 \end{bmatrix}. \tag{55}$$

For the second eigenvalue, this is

$$\begin{bmatrix} i\omega & -\omega \\ \omega & i\omega \end{bmatrix} \begin{bmatrix} x_1 \\ x_2 \end{bmatrix} = \begin{bmatrix} 0 \\ 0 \end{bmatrix} \quad \text{II} + i \cdot \text{I}$$
$$\Leftrightarrow \begin{bmatrix} i\omega & -\omega \\ 0 & 0 \end{bmatrix} \begin{bmatrix} x_1 \\ x_2 \end{bmatrix} = \begin{bmatrix} 0 \\ 0 \end{bmatrix}, \tag{56}$$

which leads to $c_2 = c$ and

$$i\omega x_1 - c\omega = 0$$
$$\Leftrightarrow x_1 = \frac{c\omega}{i\omega} \tag{57}$$
$$\Leftrightarrow x_1 = -ci$$

and therefore to

$$\bar{v}_2 = c \begin{bmatrix} -i \\ 1 \end{bmatrix}. \tag{58}$$

The complete eigenvector matrix is then

$$V = \begin{bmatrix} i & -i \\ 1 & 1 \end{bmatrix}. \tag{59}$$

### B.3.3 Inverse of eigenvector matrix

Lastly, we have to find the inverse of the eigenvector matrix

$$
\begin{aligned}
[VI] &= \begin{bmatrix} i & -i & 1 & 0 \\ 1 & 1 & 0 & 1 \end{bmatrix} \begin{matrix} \cdot -i \\ \cdot 1/2 \end{matrix} \\
&= \begin{bmatrix} 1 & -1 & -i & 0 \\ 1/2 & 1/2 & 0 & 1/2 \end{bmatrix} \mathrm{II} - {}^{1}/_{2}\mathrm{I} \\
&= \begin{bmatrix} 1 & -1 & -i & 0 \\ 0 & 1 & i/2 & 1/2 \end{bmatrix} \mathrm{I} + \mathrm{II} \\
&= \begin{bmatrix} 1 & 0 & -i/2 & 1/2 \\ 0 & 1 & i/2 & 1/2 \end{bmatrix} = [IV^{-1}].
\end{aligned}
\tag{60}
$$

### B.3.4 Solution

Now we can use the identity in equation 24 to solve the matrix exponential

$$
\begin{aligned}
\exp(At) &= V \exp(\Lambda t) V^{-1} \\
&= \begin{bmatrix} i & -i \\ 1 & 1 \end{bmatrix} \begin{bmatrix} \exp(t\lambda_1) & 0 \\ 0 & \exp(t\lambda_2) \end{bmatrix} \begin{bmatrix} -i/2 & 1/2 \\ i/2 & 1/2 \end{bmatrix} \\
&= \begin{bmatrix} i\exp(t\lambda_1) & -i\exp(t\lambda_2) \\ \exp(t\lambda_1) & \exp(t\lambda_2) \end{bmatrix} \begin{bmatrix} -i/2 & 1/2 \\ i/2 & 1/2 \end{bmatrix} \\
&= \frac{1}{2} \begin{bmatrix} \exp(t\lambda_1) + \exp(t\lambda_2) & i\exp(t\lambda_1) - i\exp(t\lambda_2) \\ -i\exp(t\lambda_1) + i\exp(t\lambda_2) & \exp(t\lambda_1) + \exp(t\lambda_2) \end{bmatrix} \\
&= \begin{bmatrix} \exp(tb)\cos(t\omega) & -\exp(tb)\sin(t\omega) \\ \exp(tb)\sin(t\omega) & \exp(tb)\cos(t\omega) \end{bmatrix}
\end{aligned}
\tag{61}
$$

to then derive the null solution

$$
\begin{aligned}
V_n(t) &= \exp(At)\bar{y}(0) \\
&= \begin{bmatrix} \exp(tb)\cos(t\omega) & -\exp(tb)\sin(t\omega) \\ \exp(tb)\sin(t\omega) & \exp(tb)\cos(t\omega) \end{bmatrix} \bar{y}(0) \\
&= \begin{bmatrix} \exp(tb)(c_1\cos(t\omega) - c_2\sin(t\omega)) \\ \exp(tb)(c_1\sin(t\omega) + c_2\cos(t\omega)) \end{bmatrix}
\end{aligned}
\tag{62}
$$

and the particular solution

$$
\begin{aligned}
V_p(t) &= \int_0^t \exp(A(t-s))\bar{q}(s)\mathrm{d}s \\
&= \int_0^t \begin{bmatrix} \exp((t-s)b)\cos((t-s)\omega) & -\exp(t-s)\sin((t-s)\omega) \\ \exp((t-s)b)\cos((t-s)\omega) & -\exp(\Delta_{ts}b)\sin(\Delta_{ts}\omega) \end{bmatrix} \\
&\qquad \begin{bmatrix} \sum_i^N w_{ij} \sum_{\{\hat{t}_i > \hat{t}_j\}} \delta(s - t_i) \\ 0 \end{bmatrix} \mathrm{d}s \\
&= \int_0^t \begin{bmatrix} \sum_i^N w_{ij} \sum_{\{\hat{t}_i > \hat{t}_j\}} \delta(s - t_i) \exp(\Delta_{ts}b)\cos(\Delta_{ts}\omega) \\ \sum_i^N w_{ij} \sum_{\{\hat{t}_i > \hat{t}_j\}} \delta(s - t_i) \exp(\Delta_{ts}b)\sin(\Delta_{ts}\omega) \end{bmatrix} \mathrm{d}s \\
&= \begin{bmatrix} \sum_i^N w_{ij} \sum_{\{\hat{t}_i > \hat{t}_j\}} \exp(\Delta\hat{t}_i b)\cos(\Delta\hat{t}_i\omega) \\ \sum_i^N w_{ij} \sum_{\{\hat{t}_i > \hat{t}_j\}} \exp(\Delta\hat{t}_i b)\sin(\Delta\hat{t}_i\omega) \end{bmatrix}.
\end{aligned}
\tag{63}
$$

The final solution is then the sum of the null and particular solutions:

$$\tilde{I}j(t) = \exp(tb)(c_1 \cos(t\omega) - c_2 \sin(t\omega)) + \sum_i^N w_{ij} \sum_{\{\hat{t}_i > \hat{t}_j\}} \exp(\Delta\hat{t}_i b) \cos(\Delta\hat{t}_i \omega) \qquad (64)$$

$$\tilde{V}_j(t) = \exp(tb)(c_1 \sin(t\omega) + c_2 \cos(t\omega)) + \sum_i^N w_{ij} \sum_{\{\hat{t}_i > \hat{t}_j\}} \exp(\Delta\hat{t}_i b) \sin(\Delta\hat{t}_i \omega). \qquad (65)$$

By setting the initial conditions $c1$ and $c2$ to 0, we arrive at the spike response model

$$\tilde{V}_j(t) = \sum_i^N w_{ij} \sum_{\{\hat{t}_i > \hat{t}_j\}} \exp(\Delta\hat{t}_i b) \sin(\Delta\hat{t}_i \omega). \qquad (66)$$

This describes only the dynamics of the subthreshold membrane potential. In contrast to the leaky integrate-and-fire neuron, all input currents are reset if an output spike is elicited and the dynamics are set to a specific point at the phase plane. Thus, in order to receive the final solution for the membrane potential, we have to add a voltage and current reset:

$$\begin{aligned}
\tilde{V}_j(t) = &\sum_i^N w_{ij} \sum_{\{\hat{t}_i > \hat{t}_j\}} \exp(\Delta\hat{t}_i b) \sin(\Delta\hat{t}_i \omega) \\
&+ \exp(\Delta\hat{t}_j b)(\tilde{V}_r \cos(\Delta\hat{t}_j \omega) + \tilde{I}_r \sin(\Delta\hat{t}_j \omega)).
\end{aligned} \qquad (67)$$

## C Log-normal probability density

Given the mean $\mu_w$ and variance $\sigma_w^2$ of a log-normal distribution [50], the cumulative distribution function is given by

$$F_X(x) = \Phi\left(\frac{ln(x) - \mu}{\sigma}\right), \qquad (68)$$

where $\Phi$ is the cumulative distribution function of the standard normal distribution and $\mu$ and $\sigma$ are the mean and standard deviation of the normal distribution underlying the log-normal distribution. The parameters $\mu$ and $\sigma$ can be derived from $\mu_w$ and $\sigma_w^2$ by solving

$$\mu = \log\left(\frac{\mu_w^2}{\sqrt{\mu_w^2 + \sigma_w^2}}\right) \qquad (69) \qquad \text{and} \qquad \sigma^2 = \log\left(1 + \frac{\sigma_w^2}{\mu_w^2}\right). \qquad (70)$$

In order to calculate the value of the $99\%$ boundary of the cumulative distribution function, we have to solve equation 68 for x

$$\begin{aligned}
0.99 &= \Phi\left(\frac{ln(x) - \mu}{\sigma}\right) \\
&\Leftrightarrow x = \exp\left(\Phi^{-1}(0.99)\sigma + \mu\right),
\end{aligned} \qquad (71)$$

where $\Phi^{-1}$ is the percent point function of the standard normal distribution. We choose the mean postsynaptic potential as a twentieth of the distance between resting potential and threshold

$$\mu_w = \frac{V_{thr} - V_0}{20} \qquad (72) \quad \text{and the variance as} \qquad \sigma_w^2 = \left(\frac{V_{thr} - V_0}{25}\right)^2. \qquad (73)$$

Inserting those values into equations 69 and 70 then yields $x \approx 0.2$, i.e. approximately $99\%$ of the cumulative distribution function is below one fifth of the distance between resting potential and threshold.

## D Normalisation factors

### D.1 Leaky integrate-and-fire neuron

The shape of the voltage kernel $K(t)$ depends on the two parameters $\tau_s$ and $\tau_m$. In the following we derive the factor $\kappa$, such that the maximum amplitude of the voltage kernel is normalised to one and therefore is exactly scaled by the size of the corresponding synaptic weights $w_{ij}$ [38, p. 143].

To this end, we first find the position of the maximum amplitude $\hat{x}$ of the voltage kernel

$$\frac{d}{dx}K(t) \overset{!}{=} 0$$

$$\Leftrightarrow \frac{d}{dx}\exp\left(-\frac{x}{\tau_m}\right) - \frac{d}{dx}\exp\left(-\frac{x}{\tau_s}\right) = 0$$

$$\Leftrightarrow \frac{1}{\tau_s}\exp\left(-\frac{x}{\tau_s}\right) - \frac{1}{\tau_m}\exp\left(-\frac{x}{\tau_m}\right) = 0$$

$$\Leftrightarrow \frac{x}{\tau_m} - \frac{x}{\tau_s} = \ln\left(\frac{1}{\tau_m}\right) - \ln\left(\frac{1}{\tau_s}\right) \tag{74}$$

$$\Leftrightarrow \frac{\tau_s x - \tau_m x}{\tau_s \tau_m} = \ln\left(\frac{\tau_s}{\tau_m}\right)$$

$$\Leftrightarrow x(\tau_s - \tau_m) = \ln\left(\frac{\tau_s}{\tau_m}\right)\tau_s \tau_m$$

$$\Rightarrow \hat{x} = \ln\left(\frac{\tau_s}{\tau_m}\right)\frac{\tau_s \tau_m}{\tau_s - \tau_m}$$

and then choose $\kappa$ such that it normalises the maximum

$$\kappa\left(\exp\left(-\frac{\hat{x}}{\tau_m}\right) - \exp\left(-\frac{\hat{x}}{\tau_s}\right)\right) \overset{!}{=} 1$$

$$\Rightarrow \kappa = \frac{1}{\exp\left(-\frac{\hat{x}}{\tau_m}\right) - \exp\left(-\frac{\hat{x}}{\tau_s}\right)}. \tag{75}$$

### D.2 Leaky resonate-and-fire neuron

The shape and amplitude of a single synaptic input

$$\exp(\Delta\hat{t}_i b)\sin(\Delta\hat{t}_i \omega) \tag{76}$$

depends on the parameters $b$ and $\omega$. In order for input spikes to cause a maximum postsynaptic potential that is equivalent to the size of the corresponding sampled weight, we normalise the maximum amplitude of synaptic inputs to one. Accordingly, we find all times at which extrema occur

$$\frac{d}{dx}\exp(xb)\sin(x\omega) \overset{!}{=} 0$$

$$\Leftrightarrow \exp(xb)(b\sin(x\omega) + \omega\cos(x\omega)) = 0$$

$$\Leftrightarrow b\sin(x\omega) = \omega\cos(x\omega) \tag{77}$$

$$\Leftrightarrow \frac{\omega}{b} = \frac{\cos(x\omega)}{\sin(x\omega)}$$

$$\Rightarrow \hat{x}(n) = \frac{\pi n - \arctan\left(\frac{\omega}{b}\right)}{\omega}, n \in \mathbb{Z}.$$

Since $b \leq 0$, $\exp(\hat{x}b)$ stays stable or decays exponentially, therefore, the biggest extremum occurs for $n = 0$. Finally, we set $\tilde{\kappa}$ such that it normalises the amplitude

$$\tilde{\kappa}\exp(\hat{x}(0)b)\sin(\hat{x}(0)\omega) \overset{!}{=} 1$$

$$\Rightarrow \tilde{\kappa} = \frac{1}{\exp(\hat{x}(0)b)\sin(\hat{x}(0)\omega)}. \tag{78}$$

## E  Event-based dynamics

The formulas for the temporal dynamics and their partial derivatives are based on sums over spike times. To avoid storing and summing over all past spike times, the formulas can be written as a recurrence relation by factorising the exponentials.

### E.1 Integrate-and-fire neuron

The equations of the membrane potential (eq. 3) and corresponding partial derivatives (eq. 8-11) of the leaky integrate and fire neuron contain sums over spike trains of the form

$$\sum_{\{\hat{t}_k\}} a \exp(b(t - \hat{t}_k)), \tag{79}$$

where $\hat{t}_k$ is either input $\hat{t}_i$ or output spikes $\hat{t}_j$, $a$ is 1 or $\Delta \hat{t}_k$ and $b$ is either $-1/\tau_s$ or $-1/\tau_m$. In the following, we first derive the recurrence relation for $a = 1$ and then analogously for for $a = \Delta \hat{t}_k$. Writing the first and second term of equation 79 with $a = 1$ and factorising the exponentials yields

$$\exp(b(t - \hat{t}_1))$$
$$\Leftrightarrow \exp(b(t - \hat{t}_1))(1) \tag{80}$$

and

$$\exp(b(t - \hat{t}_1)) + \exp(b(t - \hat{t}_2))$$
$$\Leftrightarrow \exp(b(t - \hat{t}_2))(\underbrace{\exp(b(\hat{t}_2 - \hat{t}_1)) + 1}_{c}) \tag{81}$$

Factorisation of the third term

$$\exp(b(t - \hat{t}_1)) + \exp(b(t - \hat{t}_2)) + \exp(b(t - \hat{t}_3))$$
$$\Leftrightarrow \exp(b(t - \hat{t}_3))(\exp(b(\hat{t}_3 - \hat{t}_2 - \hat{t}_1)) + \exp(b(\hat{t}_3 - \hat{t}_2)) + 1) \tag{82}$$
$$\Leftrightarrow \exp(b(t - \hat{t}_3))(c \exp(b(\hat{t}_3 - \hat{t}_2)) + 1)$$

reveals the inherent recurrent structure of the factorisation, which allows to rewrite the sum over exponentials as event-based recurrence relation

$$c_0 = 1$$
$$c_1 = c_0 \exp(b(\hat{t}_2 - \hat{t}_1)) + 1$$
$$c_2 = c_1 \exp(b(\hat{t}_3 - \hat{t}_2)) + 1 \tag{83}$$
$$c_3 = ...$$
$$\Rightarrow c^{\text{new}} = c^{\text{old}} \exp(b(\hat{t}_k - \hat{t}_{k-1})) + 1$$

and therefore the sum over exponentials as defined in eq. 79 with $a = 1$ as

$$\exp(b(t - \hat{t}_k))c_{k-1}. \tag{84}$$

Similarly, equations of the form

$$\sum_{\{\hat{t}_k\}} (t - \hat{t}_k) \exp(b(t - \hat{t}_k)) \tag{85}$$

can be separated into

$$t \sum_{\{\hat{t}_k\}} \exp(b(t - \hat{t}_k)) \tag{86}$$

and

$$\sum_{\{\hat{t}_k\}} \hat{t}_k \exp(b(t - \hat{t}_k)). \tag{87}$$

While the first part is identical to equation 79 with $a = 1$, except an additional multiplicative factor $t$, the second part can be solved by keeping track of the $\hat{t}_k$ during factorisation. Writing the first three terms yields

$$\hat{t}_1 \exp(b(t - \hat{t}_1)), \tag{88}$$

$$\exp(b(t - \hat{t}_2))(\underbrace{\hat{t}_1 \exp(b(\hat{t}_2 - \hat{t}_1)) + \hat{t}_2}_{\hat{c}}), \tag{89}$$

$$\exp(b(t - \hat{t}_3))(\hat{c}\exp(b(\hat{t}_3 - \hat{t}_2)) + \hat{t}_3), \tag{90}$$

leading to

$$\hat{c}^{\text{new}} = \hat{c}^{\text{old}}\exp(b(\hat{t}_k - \hat{t}_{k-1})) + \hat{t}_k. \tag{91}$$

In summary, this allows us to rewrite the sum over input spikes in the equation of the membrane potential (eq. 3) as

$$\Sigma_K(t) = \sum_{\{\hat{t}_i\}} K(\Delta\hat{t}_i)$$
$$= c\exp(b\Delta t_k) - c\exp(b\Delta t_k), \tag{92}$$

where the first summand uses $\hat{t}_k = \hat{t}_i$ and $b = -1/\tau_m$, and the second summand uses $\hat{t}_k = \hat{t}_i$ and $b = -1/\tau_s$. The sum over output spikes is

$$\Sigma_{\text{reset}}(t) = \sum_{\{\hat{t}_j\}} \exp\left(-\frac{\Delta\hat{t}_j}{\tau_m}\right)$$
$$= c\exp(b\Delta t_k), \tag{93}$$

with $\hat{t}_k = \hat{t}_j$ and $b = -1/\tau_m$. The sums in the derivatives with respect to the synaptic weights (eq. 8) and with respect to the reset potential (eq. 11) are also solved by $\Sigma_K(t)$ and $\Sigma_{\text{reset}}(t)$. However, the three sums in the derivatives with respect to the time constants are of different form. For the derivative with respect to the synaptic time constant it follows that

$$\Sigma_{\tau_s}(t) = \sum_{\{\hat{t}_i\}} \Delta\hat{t}_i \exp\left(-\frac{\Delta\hat{t}_i}{\tau_s}\right)$$
$$= tc\exp(b\Delta t_k) - \hat{c}\exp(b\Delta t_k) \tag{94}$$

with $\hat{t}_k = \hat{t}_i$ and $b = -1/\tau_s$ in both summands. Similarly, for the two sums in the membrane time constant

$$\Sigma_{\tau_m 1}(t) = \sum_{\{\hat{t}_i\}} \Delta\hat{t}_i \exp\left(-\frac{\Delta\hat{t}_i}{\tau_m}\right)$$
$$= tc\exp(b\Delta t_k) - \hat{c}\exp(b\Delta t_k) \tag{95}$$

with $\hat{t}_k = \hat{t}_i$ and $b = -1/\tau_m$ in both summands and

$$\Sigma_{\tau_m 2}(t) = \sum_{\{\hat{t}_j\}} \Delta\hat{t}_j \exp\left(-\frac{\Delta\hat{t}_j}{\tau_m}\right)$$
$$= tc\exp(b\Delta t_k) - \hat{c}\exp(b\Delta t_k) \tag{96}$$

with $\hat{t}_k = \hat{t}_j$ and $b = -1/\tau_m$ respectively. Please note, that the initial value for all recurrence variables is $c = 1$. This concludes the derivation of the event-dependent recurrence relations for the membrane potential of the leaky integrate-and-fire neuron and its partial derivatives.

### E.2 Resonate-and-fire neuron

The equation of the membrane potential (eq. 5) and corresponding partial derivatives (eq. 12-16) of the leaky resonate-and-fire neuron contain sums over input spike trains of the form

$$\sum_{\{\hat{t}_i > \hat{t}_j\}} a\exp(\Delta\hat{t}_i b)g(\Delta\hat{t}_i \omega), \tag{97}$$

where $a$ is either 1 or $\Delta\hat{t}_i$ and $g$ is either the sine or cosine function. In the following, we are going to derive the recurrence relation for $a = \Delta\hat{t}_i$ and the sine function, the solution for $a = 1$ and the

cosine function follow analogously. First, we note that, since synaptic currents are reset after each output spike $\{\hat{t}_i > \hat{t}_j\}$, input spike times can be written relative to the last output spike time

$$\Delta\hat{t}_i = t - \hat{t}_i = t - \hat{t}_j - (\hat{t}_i - \hat{t}_j) = \Delta\hat{t}_j - \Delta\hat{t}_{ij}. \tag{98}$$

This can be used to separate equation 97 into

$$\Delta\hat{t}_j \sum_{\{\hat{t}_i > \hat{t}_j\}} \exp(\Delta\hat{t}_i b) \sin(\Delta\hat{t}_i \omega) \tag{99}$$

and

$$\sum_{\{\hat{t}_i > \hat{t}_j\}} \Delta\hat{t}_{ij} \exp(\Delta\hat{t}_i b) \sin(\Delta\hat{t}_i \omega). \tag{100}$$

The sum in equation 99 is then again identical to equation 97 with $a = 1$. Writing the first two terms of equation 100

$$\Delta\hat{t}_{1j} \exp(\Delta\hat{t}_1 b) \sin(\Delta\hat{t}_1 \omega) + \Delta\hat{t}_{2j} \exp(\Delta\hat{t}_2 b) \sin(\Delta\hat{t}_2 \omega) \tag{101}$$

and factorising the exponentials yields

$$\exp(\Delta\hat{t}_2 b) \left[ \Delta\hat{t}_{1j} \exp((\hat{t}_2 - \hat{t}_1)b) \sin(\Delta\hat{t}_1 \omega) + \Delta\hat{t}_{2j} \sin(\Delta\hat{t}_2 \omega) \right]. \tag{102}$$

Using the identity in 98 and

$$\sin(\alpha - \beta) = \sin(\alpha)\cos(\beta) - \cos(\alpha)\sin(\beta) \tag{103}$$

this can be rewritten to

$$\exp(\Delta\hat{t}_2 b) \Bigg[ \sin(\Delta\hat{t}_j \omega) \left[ \Delta\hat{t}_{1j} \exp((\hat{t}_2 - \hat{t}_1)b) \cos(\Delta\hat{t}_{1j} \omega) + \Delta\hat{t}_{2j} \cos(\Delta\hat{t}_{2j} \omega) \right]$$
$$- \cos(\Delta\hat{t}_j \omega) \left[ \Delta\hat{t}_{1j} \exp((\hat{t}_2 - \hat{t}_1)b) \sin(\Delta\hat{t}_{1j} \omega) + \Delta\hat{t}_{2j} \sin(\Delta\hat{t}_{2j} \omega) \right] \Bigg] \tag{104}$$

which, since $\Delta\hat{t}_{ij}$ is a constant, reveals the recurrence relation

$$\exp(\Delta\hat{t}_i b) \left[ c_i \sin(\Delta\hat{t}_j \omega) - d_i \cos(\Delta\hat{t}_j \omega) \right] \tag{105}$$

with

$$\begin{aligned} c_1 &= \Delta\hat{t}_{1j} \cos(\Delta\hat{t}_{1j} \omega) \\ c_2 &= c_1 \exp((\hat{t}_2 - \hat{t}_1)b) + \Delta\hat{t}_{2j} \cos(\Delta\hat{t}_{2j} \omega) \\ c_3 &= c_2 \exp((\hat{t}_3 - \hat{t}_2)b) + \Delta\hat{t}_{3j} \cos(\Delta\hat{t}_{3j} \omega) \\ &\quad \dots \end{aligned} \tag{106}$$

and

$$\begin{aligned} d_1 &= \Delta\hat{t}_{1j} \sin(\Delta\hat{t}_{1j} \omega) \\ d_2 &= d_1 \exp((\hat{t}_2 - \hat{t}_1)b) + \Delta\hat{t}_{2j} \sin(\Delta\hat{t}_{2j} \omega) \\ d_3 &= d_2 \exp((\hat{t}_3 - \hat{t}_2)b) + \Delta\hat{t}_{3j} \sin(\Delta\hat{t}_{3j} \omega) \\ &\quad \dots \end{aligned} \tag{107}$$

Analogously, if $g$ is the cosine function, we can use

$$\cos(\alpha - \beta) = \cos(\alpha)\cos(\beta) + \sin(\alpha)\sin(\beta) \tag{108}$$

to derive the recurrence relation

$$\exp(\Delta\hat{t}_i b) \left[ c_i \cos(\Delta\hat{t}_j \omega) + d_i \sin(\Delta\hat{t}_j \omega) \right], \tag{109}$$

and finally, if $a = 1$, $c_i$ and $d_i$ are calculated without the multiplicative $\Delta\hat{t}_{ij}$ terms. More generally, we can write the recurrence relation variables as

$$c_i^{\text{new}} = c_i^{\text{old}} \exp((\hat{t}_i - \hat{t}_{i-1})b) + \hat{a} \cos(\hat{t}_i \omega) \tag{110}$$

and

$$d_i^{\text{new}} = d_i^{\text{old}} \exp((\hat{t}_i - \hat{t}_{i-1})b) + \hat{a} \sin(\hat{t}_i \omega) \tag{111}$$

with the multiplicative factor $\hat{a} \in [1, \Delta\hat{t}_{ij}]$, the arrival time of the last incoming spike $\hat{t}_{(i-1)}$ and arrival time of the current incoming spike $\hat{t}_i$.

Specifically, this allows us to rewrite the sum over input spikes in the equation of the membrane potential (eq. 5) and the partial derivative with respect to the synaptic weights (eq. 12) as

$$\begin{aligned}
\Sigma_{\tilde{V}}(t) = \Sigma_w(t) &= \sum_{\{\hat{t}_i > \hat{t}_j\}} \exp(\Delta\hat{t}_i b) \sin(\Delta\hat{t}_i \omega) \\
&= \exp(\Delta\hat{t}_i b) \left[ c_i \sin(\Delta\hat{t}_j \omega) - d_i \cos(\Delta\hat{t}_j \omega) \right],
\end{aligned} \tag{112}$$

with $\hat{a} = 1$. Further, the sum in the partial derivative with respect to the damping factor $b$ (eq. 13) with $\hat{a} = \Delta\hat{t}_j$ is

$$\begin{aligned}
\Sigma_b(t) &= \sum_{\{\hat{t}_i > \hat{t}_j\}} \Delta\hat{t}_i \exp(\Delta\hat{t}_i b) \sin(\Delta\hat{t}_i \omega) \\
&= \Delta\hat{t}_j \Sigma_{\tilde{V}}(t) - \exp(\Delta\hat{t}_i b) \left[ c_i \sin(\Delta\hat{t}_j \omega) - d_i \cos(\Delta\hat{t}_j \omega) \right].
\end{aligned} \tag{113}$$

And finally, we can rewrite the sum in the partial derivative with respect to the frequency of the subthreshold oscillations $\omega$ (eq. 14) as

$$\begin{aligned}
\Sigma_\omega(t) &= \sum_{\{\hat{t}_i > \hat{t}_j\}} \Delta\hat{t}_i \exp(\Delta\hat{t}_i b) \cos(\Delta\hat{t}_i \omega) \\
&= \Delta\hat{t}_j \exp(\Delta\hat{t}_i b) \left[ c_i \cos(\Delta\hat{t}_j \omega) + d_i \sin(\Delta\hat{t}_j \omega) \right] \\
&\quad - \exp(\Delta\hat{t}_i b) \left[ c_i \cos(\Delta\hat{t}_j \omega) + d_i \sin(\Delta\hat{t}_j \omega) \right],
\end{aligned} \tag{114}$$

where the first pair of recurrence variables $c_1$ and $d_1$ use $\hat{a} = 1$ and the second pair $\hat{a} = \Delta\hat{t}_{ij}$. This concludes the derivation of the event-based update rules.

# F    Learning dynamics contract and do not converged

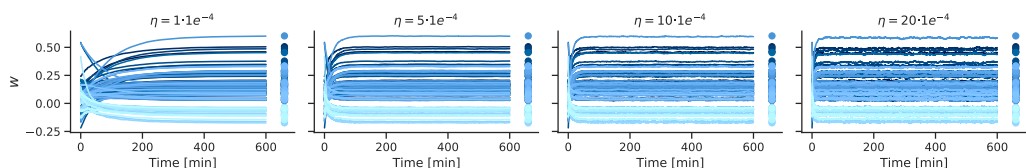

Figure 5: Training the synaptic weights of a LRF neuron with stochastic gradient descent reveals that the learning dynamics (eq. 1) contract and do not converge to the solution. Parameters get close to but then keep fluctuating around their target value. The size of the fluctuations is dependent on the size of the learning rate $\eta$.

# G    Varying target computations

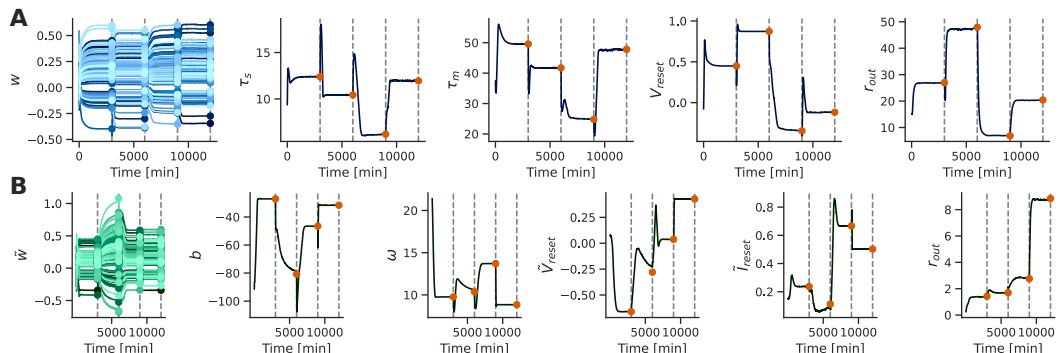

Figure 6: The online learning algorithm can adapt to varying target computations. A, B: If the target computation is changed (dotted vertical lines), the student smoothly adapts and converges towards the new target computation (dots) in LIF (A) and LRF (B) neurons.

# H    Training synaptic weights only

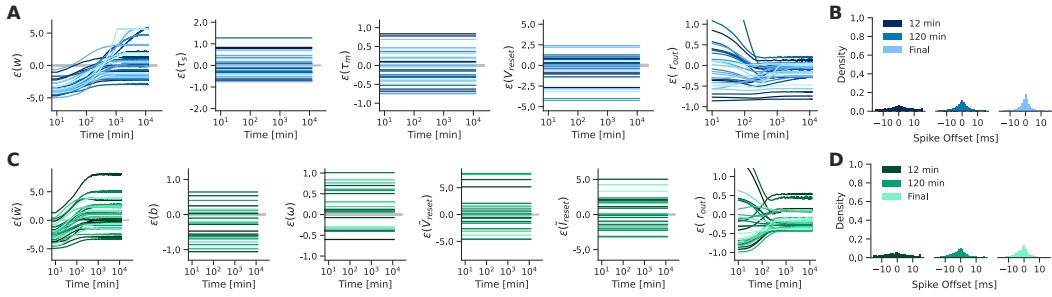

Figure 7: Solely training synaptic weights is insufficient to learn target computations. A, C: When sampling initial parameters as defined in section 2.5 and subsequently optimising synaptic weights while keeping other parameters fixed, synaptic weights and output-firing rates do not converge to their target values in LIF (A) and LRF (C) neurons. B, D: As training progresses from $0.1\%$, $1\%$ to $100\%$ of training time, the probability distribution of students' output spike times around target spikes for LIF (C) and LRF (D) neurons maintains high variance i.e. fails to accurately map input spike trains to target spike trains (cf. Fig 4G,H).

# I Vanilla training

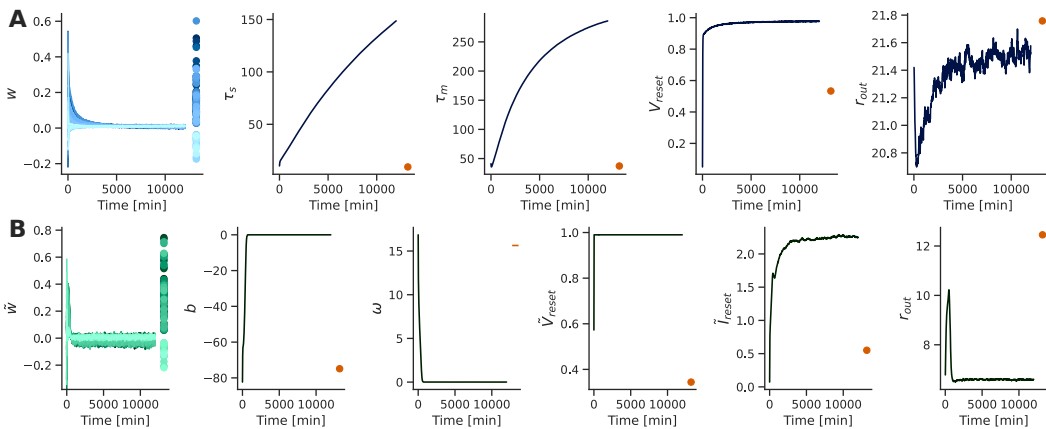

Figure 8: Online learning without the event-dependent scaling factor fails. A, B: When training a LIF (A) or LRF (B) neuron according to the gradient-based learning rule (eq. 1) wihtout the scaling-factor $\lambda(\Delta d^{\pm})$, the synaptic weights collapse and the intrinsic parameters diverge. Interestingly, the LIF neuron's target firing rate $r_{\text{out}}$ is getting close to the target value (dot) despite the diverging parameters.

# J LRF parameter combinations

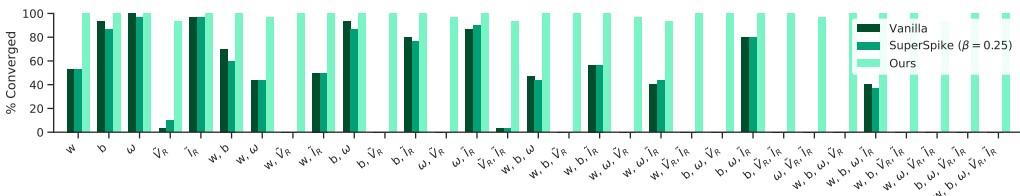

Figure 9: Percentage of $n = 30$ students that converged to the target values for all possible parameter combinations with vanilla, surrogate gradient and the EDS algorithm in LRF neurons. Surrogate gradients performs similar to the vanilla algorithm. EDS reliably contracts for all parameter combinations and in particular outperforms the other two algorithms for parameter combinations that include the voltage reset $V_r$.

## K    Surrogate gradient learning trajectories

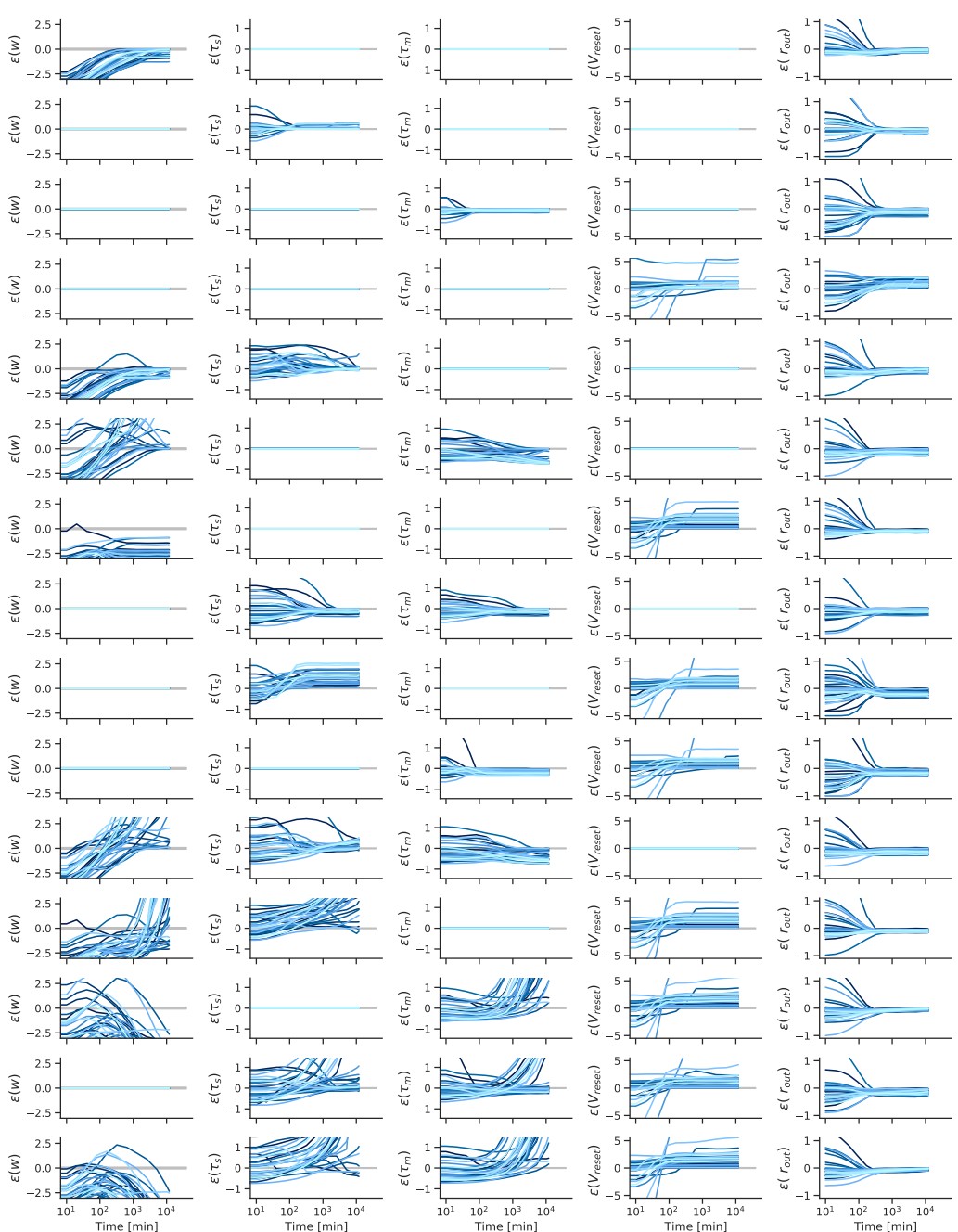

Figure 10: Learning trajectories for LIF neurons with SuperSpike surrogate gradient ($\beta = 0.25$), for all possible parameter combinations. Even when trained only on synaptic weights (first row), many initialisation get stuck in a local minimum with a close to accurate output firing rate but synaptic weights that are different from the teacher's weights, i.e. a different input-output function. Similarly, for other parameter combinations, the output firing rate converges to and is kept close to the right level, however, often at the cost of inaccurate or even diverging parameter values.