# OpenReview forum: "Online Learning Of Neural Computations From Sparse Temporal Feedback"
_NeurIPS.cc/2021/Conference — NeurIPS 2021 Poster_

### Official Review · Reviewer_meDX · 2021-07-15

**Rating:** 7
**Confidence:** 3

**Summary:**

This paper introduces the event-dependent scaling (EDS) rule to use in neural network training that allows neural computations to be learned online - i.e. without error propagation. It presents a teacher-student paradigm for problem formulation, then characterizes the error signal and EDS rule. It then describes its architectures (leaky integrate-and-fire or leaky resonate-and-fire neurons, Poisson-distributed input spike trains, and other standard parameters), and metrics (of convergence and robustness). Next, it discusses resulting learning dynamics and analyses the effectiveness of EDS empirically, and concludes by acknowledging concerns behind lack of theoretical backing and a rarefied testing paradigm.


**Limitations And Societal Impact:**

The limitations are discussed in detail and hit upon all concerns I could think of. My only disagreements with the authors would be on the magnitude of the limitations, not the limitations themselves. The authors don’t really discuss societal impact, and there may be some from accurate models of biological intelligence that can’t be dismissed by saying the work is theoretical, but we are nowhere close so I don’t think it warrants a flag.


**Main Review:**

*Originality:* The primary contribution of this paper is an event-dependent scaling online learning rule. To my knowledge, this is a novel and original contribution to the field, building on top of existing work in online learning rules, spiking architectures, and methods of training spiking architectures given that they are nondifferentiable. The introduction cites helpful relevant work to set up the problem, and the related work subsection does both careful and thorough review of existing work in related fields and differentiation from similar work.

*Quality:* I believe this paper is technically sound. The empirical analysis is convincing and the figures are helpful - I don’t feel that the text overstates the effectiveness of the EDS rule. The ablation and gradient analysis in section 3.4 is useful, and reasonable and accurate to the best of my understanding. The dynamics in Fig 3 and various descriptive statistics shown in Fig 4 are effective and say a lot on their own.

The methods are also detailed and make a lot of sense. I think section 2 is the best section of the paper - it lays out each section of the experimental setup well, highlights the contribution in section 2.3, and keeps all non-novel parts standard and familiar. My only concern with methods is the teacher-student paradigm. It feels very toy because the solution is both single-dimensional and an exact imitation of the teacher. This could easily be a limitation of my own knowledge, but the uses of teacher-student I’ve seen before (and the ones discussed in this paper) are from a very long time ago and generally used for theoretical work where the simplicity of the problem is at least mitigated by the fact that the solution is provably reachable/entailed by the initialization. For an empirical work, it feels like a good but not independent contribution, but again I recognize that this could be a limitation of my knowledge. I will say that the authors discuss this limitation and others with honesty and detail, and the paper seems to warrant its claims.

*Clarity:* The clarity of this paper fluctuates. Section 1 is clearly written, and section 2 is not only clearly written but well-organized, with each subsection answering the question I have or the next thing I want to know at the end of the previous section. I think this is also true of the subsection structure in section 3.

The writing in Section 3 is less clear, and sounds more like a process statement than results and analysis. There is a lot of “we designed X”, “we trained Y”, etc. This is mainly a stylistic issue, but it does make the organization feel temporal/chronological rather than claims-driven and this leads to a results section that is hard to understand despite holding valuable information. It would help to have some framework at the beginning that outlines the flow of the argument in the section 3 subsections and makes all main claims, let each subsection derive from that framework.

*Significance:* I find this method to be important and unique. The results seem to open a new direction in online learning without backprop that people could in theory expand or extend upon. Insofar as spiking neurons are a good model of the brain, this paper seems to advance their modeling capacity. However, I am not confident that the expansion or extension is actually viable. The problem seems very toy and while it could be a limitation of my own knowledge, I don't know it to be one that has a history of transferring well to more complex domains.


**Time Spent Reviewing:**

6

---

> ### Author Response · Authors · 2021-08-10
> **Answer To Review By Reviewer meDX**
>
> We thank the reviewer for the positive comments and thorough review.
>
> __1. “My only concern with methods is the teacher-student paradigm. [...] The problem seems very toy and while it could be a limitation of my own knowledge, I don't know it to be one that has a history of transferring well to more complex domains."__
>
> Their concern over the teacher-student paradigm to be “a rarefied testing paradigm” is understandable, but we believe that it’s an important first step to validate our algorithm that opens the door to a much wider field of investigation. The teacher-student paradigm (e.g. used in [29] (see “Reconstruction of Synaptic Weights”)) is a special case of the two most frequent paradigms used to evaluate learning algorithms for spiking neurons. The first paradigm (e.g. used in [12, 13, 23]) tests the ability of spiking neurons to map randomly sampled input spike patterns to random output spike patterns. The second trains neurons to distinguish between two independent sets of fixed randomly sampled input spike patterns (e.g. [9, 14, 19, 20]). These two paradigms are also used to compare different algorithms (e.g. in Wang, Xiangwen, Xianghong Lin, and Xiaochao Dang. "Supervised learning in spiking neural networks: A review of algorithms and evaluations." Neural Networks 125 (2020): 258-280.). They can be considered as training paradigms which test an algorithm’s general capability of performing spike-pattern recognition.
>
> In contrast to the first paradigm, the teacher-student paradigm generates target spike times using a model (i.e. the teacher neuron), rather than using randomly sampled target spike times. This approach has the advantage that a solution is guaranteed to exist and that it is a single point in parameter space (i.e. the task is solvable but only by one set of parameters). This is in stark contrast to random input-output spike mappings in which the difficulty of the task is directly dependent on the dimensionality of the input, the statistics of the input spike patterns (e.g. firing rate) and the amount of target spikes. Under the original, first paradigm, the size and shape of the solution space and therefore the difficulty of the task are unknown and strongly fluctuate with the choice of hyperparameters. Further, using random input-output spike patterns of fixed length is unsuitable for evaluating an online-learning algorithm because it necessarily requires replaying patterns since, as the length of the random pattern increases, the probability of a solution existing vanishes to 0.
>
> In summary, we argue that the teacher-student paradigm is an improved, and  more well-defined learning problem to validate an algorithm’s capability of performing spike-pattern recognition in spiking neuron models than the more commonly used random input-output patterns. Therefore, our simulation studies provide strong evidence that if a relation between input-spike patterns and output spikes exists (even if the solution space is only a single point in parameter space) that the Event-Dependent Scaling rule presented in this work can find it online and even when training signals are perturbed by noise (Fig. 4G, 4H) or the underlying target computation is changing over time (additional Fig. G). This is a strong indicator of the algorithm’s capability of performing general spike pattern recognition. We will raise these points more clearly in the updated manuscript.
>
> If the reviewer has a suggestion for a “more complex domain” to test our algorithm’s learning ability for spike pattern recognition we would be keen to investigate and possibly add an additional figure.
>
> __2. “acknowledging concerns behind lack of theoretical backing”__
>
> Regarding the theoretical backing of the algorithm, we ask the Reviewer to refer to point 3 in our response to Reviewer 2.
>
> __3. "The writing in Section 3 is less clear, and sounds more like a process statement than results and analysis. There is a lot of “we designed X”, “we trained Y”, etc. This is mainly a stylistic issue, but it does make the organization feel temporal/chronological rather than claims-driven and this leads to a results section that is hard to understand despite holding valuable information. It would help to have some framework at the beginning that outlines the flow of the argument in the section 3 subsections and makes all main claims, let each subsection derive from that framework."__
>
> The reviewer remarks that the paper could benefit from restructuring the results (section 3) from its current form by adding an introduction which addresses the main claims before going into details. We agree that an introduction (and removing the stylistic issues mentioned) will improve the readability of the results section and are happy to address these issues in the final version of the paper.

---

> > ### Comment · Reviewer_meDX · 2021-08-17
> > **Changing to accept**
> >
> > I was originally concerned about the lack of generalization capability of the teacher-student paradigm, and this is what my marginal reject was based on. However, seeing that other reviewers are less concerned and, crucially, that this explanation of the teacher-student paradigm adds a lot to my knowledge and is convincing, I am convinced that my concerns were less founded than I originally thought.
> >
> > Also appreciate the positive response to my feedback on clarity!
> >
> > Thanks for the detailed response!

---

### Official Review · Reviewer_V7xW · 2021-07-15

**Rating:** 7
**Confidence:** 2

**Summary:**

This work demonstrates that the intrinsic parameters of a leaky integrate-and-fire neuron and a resonate-and-fire neuron can be learned in a student-teacher paradigm. The student neuron learns the intrinsic parameters of the teacher neuron using only the teacher neuron's spike times and gradient based updates.

**Ethical Concerns:**

I have no ethical concerns.

**Limitations And Societal Impact:**

The authors address the fact that the EDS rule lacks a theoretical justification; however, the other weaknesses listed above are not addressed.

**Main Review:**

Strengths:

1. This work demonstrates that the internal parameters of a neuron can be learned using only spike-time feedback. This is important since the neural computations are sensitive to changes in intrinsic parameters. While most ANN models focus on synaptic updates, this work suggests that adapting the internal parameters of a neuron can play an important rule in learning.

Weaknesses:

1. The work seems like an extension of [29] to include internal parameters of a neuron and therefore the results are perhaps not surprising.

2. If the reader is to be convinced that the internal parameters of the neuron are important for learning, then there should be a comparison of the expressivity of a neuron with fixed internal parameters (but adaptive synapses) and a neuron with adaptive internal parameters (and adaptive synapses).

3. The EDS rule is presented without a theoretical justification.

Overall: This work is an interesting proof-of-concept demonstration that the internal parameters of a neuron can be learned and suggests that these parameters should be given more attention.

**Time Spent Reviewing:**

3

---

> ### Author Response · Authors · 2021-08-10
> **Answer To Review By Reviewer V7xW**
>
> We thank the reviewer for their thoughtful critique of our manuscript.
>
> __1. “Only extension of [29] … results perhaps not surprising”__
>
> The reviewer is concerned that our work is a somewhat “unsurprising” application of Memmesheimer et al., [29] to intrinsic parameters. We would have to politely disagree. Of course we built on the ideas of [29], developing a gradient-based online learning algorithm from a loss function which only indicates true and false positive spikes. However, we present a new learning rule which includes an event-dependent scaling factor, allowing us to do full online learning (new) of synaptic weights and intrinsic parameters (new) in both LIF and LRF (new) neurons, and without (previously used) additional constraints such as a full reset of time after each error, addition of a temporal window around target spike times in which outputs are considered to be correct and assuming a training signal free from temporal noise. The event-dependent learning rule is a main contribution of our paper that allows for full mutual online learning of synaptic weights and intrinsic parameters. We will make these advances more clear in lines 66-70.
>
> __2. “there should be a comparison of the expressivity of a neuron with fixed internal parameters (but adaptive synapses) and a neuron with adaptive internal parameters (and adaptive synapses)”__
>
> We very much appreciate the reviewers note that a comparison between learning synaptic weights only in contrast to learning intrinsic parameters and synaptic weights mutually is missing. We ran an additional simulation (and will include a corresponding plot in the revised supplementary material), which compares the spike offsets (as in figure 3C and 3D) between models that optimise synaptic weights only, and models that optimise both, synaptic weights and intrinsic parameters for LIF and LRF neurons. To this end, we sampled initial parameters as defined in section 2.5 but subsequently optimised synaptic weights only. The simulations show that when optimising synaptic weights only, the final synaptic weights deviate from the teacher’s values and only 18.5% + 0.7% SE (12.9% + 0.6% SE) of output spikes are direct hits and an additional 25.3% (19.8%) fall within the 1ms window around the target spike time. In stark contrast to the >99% learned spikes for the model with intrinsic parameter optimisation (section 3.3, Fig. 3C and 3D), this only addresses <44% and <33% of total spikes for LIF and LRF neurons respectively.
>
> __3. No theoretical justification__
>
> In agreement with the reviewer’s comment we had already flagged the lack of a formal proof of convergence in our manuscript, but we would like to stress that section 3.2 and figures 4E and 4F respectively give very strong insight of a) why online-learning and matching output spike times in contrast to firing rates is difficult (output and target spikes become more and more correlated which results in increasingly frequent wrong vanilla gradients) and b) how the event-dependent-scaling factor helps to resolve this issue (rescaling gradients such that they point into the right direction). This level of functional understanding is way beyond what we currently have for the state of the art surrogate gradient methods, for which, to the best of our knowledge, a theoretical explanation is still lacking, as well.

---

### Official Review · Reviewer_awD2 · 2021-07-16

**Rating:** 7
**Confidence:** 3

**Summary:**

The current paper presented an algorithm using sparse feedback signals at spikes and gradient-based updating rule to train the intrinsic parameter of spiking neuron models. The authors laid out the parameter updating rules for leaky-integrate-and-fire neuron and resonant-and-fire neuron models, and demonstrated the algorithm in training the two neuron types to recover a teacher signal.

**Limitations And Societal Impact:**

The authors acknowledged that the current parameter updating rule lacks theoretical support. In addition, the current work demonstrated the learning algorithm in a simple teacher-student training paradigm. It is unclear how it would behavior in a more complicated training environment.

**Main Review:**

In general, the current work presented a learning algorithm for intrinsic properties of spiking neurons. Following the current learning rule, the parameter update rate was both modulated by the sign of an event (a miss or false positive spike), and also regulated by the proximity of the event. The authors showed that the current parameter updating rule allows various parameters to converge robustly. In contrast, a ‘vanilla’ updating rule (without temporal dependency) or a surrogate rule (instead of temporal dependency, it use voltage dependency rule) both were failed to converge in multiple simulations. The paper was well written and messages were clearly delivered. However, the learning algorithm converges extremely slow on neurons with relatively low firing rate (<10 Hz, >10 h). Notice that this range of firing rate (< 10 Hz) is biologically meaningful. The slow convergence might limited the application of the current algorithm.

**Time Spent Reviewing:**

5

---

> ### Author Response · Authors · 2021-08-10
> **Answer To Review By Reviewer awD2**
>
> We thank the reviewer for their thoughtful reading of our manuscript and the constructive feedback.
>
> __1. “Converges extremely slowly … for low firing rate (<10 Hz, >10 h) … but (< 10Hz) biologically meaningful … might limit(s) the application of the current algorithm”__
>
> The reviewer noted that in the biological meaningful regime of firing rates <10Hz, the learning algorithm converges very slowly (>10h) and raised concerns regarding the applicability of the algorithm. It is not immediately clear if the reviewer was concerned about the applicability for computational modelling or as a biological mechanism,  in the following we try to address both.
>
> Firstly, we would like to note that we used a very conservative convergence criterion (section 2.6, resulting in >99% of spikes to fall within 1ms of target spike times at a temporal simulation resolution of 1ms). Training speed could be increased dramatically (>10x) by increasing learning rates, albeit at the cost of losing precision (see additional figure F). To avoid such loss in precision, one could use learning rate scheduling to speed up initial learning and subsequently gain precision as the learning rate decays. Convergence time could be further reduced by using batch-learning, replaying training data instead of using strict online learning, and by using educated guesses for initial model parameters when modeling biological data (e.g. initialise parameters such that the initial firing rate of the data and model are identical). We only used a fixed and “slow” learning rate in order to be able to adapt to varying target computations (see additional figure G) while achieving high precision at the same time, without complicating the algorithm with an additional learning rate schedule. We will note these points in a future version of the manuscript.
>
> From a biological perspective, we would like to note that most electrophysiological changes induced by plasticity mechanisms are gradual and a complete reorganization of a neuron’s electrophysiology (like what we have shown) is implausible. As a proof of principle, we started each simulation at a random point in parameter space and required the algorithm to converge to a second, randomly sampled point in parameter space. This could lead to long learning trajectories and thus to slow convergence times, especially when firing rates were low. In biologically more plausible scenarios, the start and end point are likely closer.
>
> Regarding the applicability in computational modelling, we would like to emphasise that we always refer to simulated time in our manuscript, not real time. Real time, i.e. the time required to run a simulation was always several orders of magnitude faster than simulated time. For example, 30 independent simulations of 12,000 minutes of simulated time at 1ms resolution (as in Fig. 3), take less than 30 minutes of real time on a single AMD Ryzen 5950x. We apologise if this was a source of confusion and will make a note in the updated version of the manuscript.
>
> In summary, our convergence times can be understood as an upper bound that resulted from the strict online learning assumption we made (learning from a single continuous stream of information, without batch-learning, replaying data or resetting the neuron’s state), from reducing the task to a single point in parameter space via the teacher-student paradigm, and from the choice of a conservative convergence criterion. By weakening any of these assumptions or by using tricks available from the vast toolbox of gradient-descent optimisation, convergence times could be drastically reduced, leading to faster simulation times.
>
> __2. “Unclear how it would behave in a more complicated environment”__
>
> The reviewer also raised concerns that the teacher-student paradigm may not allow extrapolation to “more complicated environments”. - We would kindly refer to the discussion with reviewer number 3 below. As an aside, if the reviewer would like to suggest additional environments to evaluate  online-learning algorithms for spiking neurons we would be keen to try, and provide an additional figure.

---

### Decision · Program_Chairs · 2021-09-27

**Decision:**

Accept (Poster)

**Comment:**

This paper presents an approach for optimizing the intrinsic parameters of spiking neurons, and tests it within a teacher-student paradigm. The initial reviews were generally positive, and reviewers agreed the paper was technically sound and interesting, though there was some concern about the paradigm used for evaluation. However, after the authors' rebuttals, and some discussion amongst reviewers, a consensus was reached that this paper merits acceptance at NeurIPS.